# Vector characteristics of the sound field based on a single vector hydrophone

Chao Wang[1,2,3], Shuyang Jia[1]*, Yanhou Zhang[1], Qi Zhang[1], Rongxin Zhu[1]

**1** Navy Submarine Academy, Qingdao, Shandong, China, **2** Lao Shan Laboratory, Qingdao, Shandong, China, **3** Qingdao Institute of Collaborative Innovation, Qingdao, Shandong, China

\* 18702512077@163.com

## Abstract

The acoustic field in the ocean is a vector field, and its vector characteristics can be gained by a single vector hydrophone. This paper focuses on the vector characteristics of the sonic field, analyzes the coherence function, energy component proportion, and phase spectrum of the received signal from a single vector hydrophone. The purpose of the study is to establish a research foundation for applying the vector characteristics of sound field to target detection and recognition. The simulation results demonstrate that the total coherence function of the sound field is approximately 0.5 when the signal-to-noise ratio is −10 dB. At this point, the coherence spectrum of the sound field is approximately equivalent to the diffusion spectrum, and the target quadrant can be identified based on the phase difference of the sound field. The results of the anechoic pool and sea testing indicate that under the background noise condition, the coherence function values are generally below 0.2, except for some specific frequency points. Using coherence spectrum for detection can improve the SNR by about 8 dB. The overall coherence function of the sound field, the total sonic field spectrum and the coherence spectrum show evident interference of moving targets.

## Introduction

Sonic wave is a form of propagation of mechanical vibration of a medium particle in a continuous media from near to far. The space where sound waves exist is called an acoustic field. The sonic field is a vector field that contains the sound pressure scalar information, particle vibration speed, acceleration and other vector information. If only the sound pressure scalar is measured, many phenomena within it cannot be observed. The vector characteristics of the sound field can only be obtained by measuring the three orthogonal components of the scalar sound pressure and the particle vibration velocity, or the particle vibration acceleration vector, in the sound field simultaneously [1–4]. Compared with the traditional sound pressure hydrophone, the vector hydrophone has remarkable advantages: 1) In

**Data availability statement:** All relevant data for this study are publicly available from the figshare repository (https://doi.org/10.6084/m9.figshare.31483549).

**Funding:** This work was supported by National Key R&D Program of China (Grant No. 2021YFC3100900), Laoshan Laboratory Science and Technology Innovation Project (Grant No. LSKJ202201100, LSKJ202500300, LSKJ202500900), and The Innovation Plan of Qingdao Institute of Collaborative Innovation (Grant No. LYY-2022-05). All funds were received by Chao Wang. The funders above provide financial support only and no role in study design.

**Competing interests:** The authors have declared that no competing interests exist.

the observation of sound field information [5–9], the vector hydrophone can synchronously measure the sound pressure and particle velocity, thereby providing more comprehensive sound field information and effectively broadening the space of signal processing. 2) In terms of target detection [10–16], the velocity channel of the vector hydrophone exhibits a frequency-independent dipole directivity and a certain ability to resist isotropic noise. Using a single vector hydrophone can achieve unambiguous direction finding throughout the entire space of the target, which gives the vector hydrophone unique advantages in the field of target detection. 3) When it comes to carrying platforms [17–23], vector hydrophone offers a smaller size, lighter weight and lower power consumption compared with acoustic pressure hydrophone array, especially in low-frequency detection. It can effectively overcome limitations in energy and size faced by integrated acoustic detection of underwater unmanned platforms.

A single vector hydrophone acquires two fundamental physical quantities in the sound field: sound pressure and particle vibration velocity, and then derives the vector characteristics. Furthermore, fully utilizing the distinct vector characteristics enables target detection and identification. At present, the target direction-finding algorithm based on a single vector hydrophone has been extensively studied by domestic and foreign scholars [24–28]. However, the vector characteristics of the sound field are relatively limited. In this paper, the coherence function, energy component, and phase spectrum of the received signal from a single vector hydrophone are analyzed in detail. The above three vector characteristics of the sound field were discussed and analyzed by computer simulation, anechoic pool testing, shallow sea testing and testing in the South China Sea. The results show that vector characteristics such as coherence function, energy component proportion, and phase spectrum of sonic field can be utilized in the field of target detection.

**Conventional single-point coherence function and phase spectrum theory**

The concepts of conventional single-point coherence function and phase spectrum are defined in ocean vector acoustics. First, the coherence function describes the correlation of two random processes in the frequency domain. It represents the square of the normalized sound intensity at a given frequency in the sound field. Compared with the correlation function, the coherence function provides more information. It reflects the proportion of coherent components and diffuse components (non-coherent components) in the total energy of the background noise field. Second, the phase spectrum reflects the distribution of the phase difference of the two components in the frequency domain. Besides, it is a crucial physical quantity that reflects the characteristics of the noise vector field. The calculation formula for the conventional single-point coherence function and phase spectrum in the sound field is:

$$\gamma_{PV_i}^2\left(f\right) = \frac{\left|\langle S_{PV_i}\left(f\right)\rangle\right|^2}{\langle\left|S_p\left(f\right)\right|^2\rangle \cdot \langle\left|S_{V_i}\left(f\right)\right|^2\rangle} \quad (i = x, y, z)$$

(1)

$$\Delta\varphi_{PV_i} = \arctan\left[\frac{\mathrm{Im}\left(S_{PV_i}(f)\right)}{\mathrm{Re}\left(S_{PV_i}(f)\right)}\right] \quad (i = x, y, z)$$

(2)

$S_p(f)$ and $S_{V_i}(f)$ $(i = x, y, z)$ respectively represent the FFT frequency domain results of sound pressure $p(t)$ and particle vibration velocity $v_i(t)$ in each direction. $S_{PV_i}(f)$ denotes the product of the complex conjugation of sound pressure and particle velocity. "Im" expresses the complex imaginary part, while "Re" signifies the complex real part. The statistical average in time is represented as $\langle\cdot\rangle$. The value range of the coherence function $\gamma^2_{PV_i}(f)$ is $[0\ 1]$. When the coherence function $\gamma^2_{PV_i}(f) = 0$, the signal component $p(t)$ and $v_i(t)$ is completely non-coherent; When the coherence function $\gamma^2_{PV_i}(f) = 1$, thus $p(t)$ and $v_i(t)$ is completely coherent; If $0 < \gamma^2_{PV_i}(f) < 1$, there are both coherent and non-coherent components between $p(t)$ and $v_i(t)$. The coherence function characterizes the energy relationship in the sound field, indicating the proportion of the coherent component and the diffused component in the total energy of the acoustic field.

The phase spectrum reflects the distribution of the phase difference between the two components in the frequency domain. It contains the supplementary information of the coherence function, which is a significant physical quantity expressing the characteristics of the vector field. For the three-dimensional vector hydrophone, if the sound propagation direction aligns with the positive direction of the three orthogonal axes of the particle speed, the phase difference between the sound pressure and the particle vibration speed is 0; if they are opposed, the phase difference will be equal to 180°. The phase difference between the sound pressure and the three-channel particle vibration velocity is as follows:

$$\Delta\varphi_{PV_x} = \begin{cases} 0 & (270^\circ < \theta < 90^\circ) \\ \pi & (90^\circ < \theta < 270^\circ) \end{cases}$$
$$\Delta\varphi_{PV_y} = \begin{cases} 0 & (0^\circ < \theta < 180^\circ) \\ \pi & (180^\circ < \theta < 360^\circ) \end{cases}$$
$$\Delta\varphi_{PV_z} = \begin{cases} 0 & (0^\circ < \phi < 90^\circ) \\ \pi & (90^\circ < \phi < 180^\circ) \end{cases}$$

(3)

$\theta \in [0, 2\pi]$ is the azimuth angle of the arrival signal (the angle between the incident sound wave and the positive direction of the X axis, and the target rotates counterclockwise from X-axis positive direction for a period of 0 °~360 °). $\phi \in [0, \pi]$ denotes the pitch angle of the arrival signal (the angle between the incident sonic wave and the positive direction of the Z-axis). Formula (1) defines the conventional single-point coherence function among three orthogonal components of sound pressure and particle velocity. The overall coherence function of the sound field is [7]:

$$\gamma^2_{PV}(f) = \frac{\sum_i \left|\langle S_{PV_i}(f)\rangle\right|^2}{\langle|S_p(f)|^2\rangle \cdot \sum_i \langle|S_{V_i}(f)|^2\rangle} \quad (i = x, y, z)$$

(4)

In spherical coordinates, the relationship between the three components of vibration velocity and the total vibration velocity can be expressed as $v_x = V\sin\theta\cos\phi$, $v_y = V\sin\theta\sin\phi$, $v_z = V\cos\theta$. Therefore, the derivation is as follows:

$$\gamma^2_{PV_i}(f) = \frac{\left|\langle S_{PV_i}(f)\rangle\right|^2}{\langle|S_p(f)|^2\rangle \cdot \langle|S_{V_i}(f)|^2\rangle} \quad (i = x, y, z)$$

(5)

$$\gamma_{PV}^2(f) = \frac{\sum_i \left|\langle S_{PV_i}(f)\rangle\right|^2}{\langle |S_p(f)|^2\rangle \cdot \sum_i \langle |S_{V_i}(f)|^2\rangle} \quad (i = x, y, z)$$

$$= \frac{\left|\langle S_{PV_x}(f)\rangle\right|^2 + \left|\langle S_{PV_y}(f)\rangle\right|^2 + \left|\langle S_{PV_z}(f)\rangle\right|^2}{\langle |S_p(f)|^2\rangle \cdot \left(\langle |S_{V_x}(f)|^2\rangle + \langle |S_{V_y}(f)|^2\rangle + \langle |S_{V_z}(f)|^2\rangle\right)}$$

$$= \frac{\left|\langle S_{PV}(f)\rangle\right|^2 \cos^2\theta\cos^2\phi + \left|\langle S_{PV}(f)\rangle\right|^2 \cos^2\theta\sin^2\phi + \left|\langle S_{PV}(f)\rangle\right|^2 \sin^2\theta}{\langle |S_p(f)|^2\rangle \cdot \left(\langle |S_V(f)|^2\rangle\cos^2\theta\cos^2\phi + \langle |S_V(f)|^2\rangle\cos^2\theta\sin^2\phi + \langle |S_V(f)|^2\rangle\sin^2\theta\right)}$$

$$= \frac{\left|\langle S_{PV}(f)\rangle\right|^2}{\langle |S_p(f)|^2\rangle \cdot \langle |S_V(f)|^2\rangle} \tag{6}$$

The energy in the sound field consists of two parts: coherent components and non-coherent components (diffusion components).

The coherent component of the sound field arises from the superposition of phase-correlated multipath signals (such as direct arrivals via fixed paths and specular reflections from the sea surface and seafloor), while the diffuse component stems from signals with random phase fluctuations (such as rough boundary scattering and perturbation scattering in the seawater medium). We explicitly assume that the phase fluctuations of the coherent and diffuse components are mutually independent random processes, and that the amplitude fluctuations of the two components exhibit no statistical correlation.

Starting from the expression for the complex sound pressure of the acoustic field, the total sound pressure can be decomposed into the coherent component $p_c$ and the diffuse component $p_d$, i.e., $p = p_c + p_d$.

The time-averaged total sound intensity is given by:

$$\langle I\rangle = \left\langle |p|^2\right\rangle = \left\langle |p_c|^2\right\rangle + \left\langle |p_d|^2\right\rangle + \left\langle p_c * p_d^* + p_c^* p_d\right\rangle \tag{7}$$

Based on the assumption of statistical independence between the coherent and diffuse components, the time average of the cross terms satisfies:

$$\left\langle p_c * p_d^* + p_c^* p_d\right\rangle = \langle p_c\rangle\langle p_d^*\rangle + \langle p_c^*\rangle\langle p_d\rangle = 0 \tag{8}$$

The physical essence is that the phase of the diffuse component is randomly distributed and, under time averaging, uncorrelated with the fixed phase of the coherent component, causing the cross-term contributions to cancel each other out. Therefore, the energy decomposition formula $I = I_c + I_d$ holds.

Here, we further discuss the applicability of this assumption. The validity of this assumption depends on two prerequisite conditions; beyond these, the accuracy of the energy decomposition will decrease:

The phase fluctuations of the diffuse component satisfy ergodicity, and their fluctuation scale is much smaller than the phase stability time of the coherent component. In shallow-sea environments with high signal-to-noise ratios and moderate boundary roughness, this condition is well satisfied. However, if the boundary is extremely rough (e.g., a gravel seabed), the proportion of the diffuse component becomes too high and partial correlation appears between its phase fluctuations and those of the coherent component, rendering the cross terms non-negligible.

The random inhomogeneity of the seawater medium constitutes a weak perturbation. If strong turbulence causes drastic fluctuations in the refractive index, the phase of the coherent component will be disturbed, thereby violating the statistical independence from the diffuse component.

The coherence function represents the ratio of these two components in the total energy of the sonic field as follows:

$$S_{tot}(f) = S_{coh}(f) + S_{dif}(f) \tag{9}$$

 

$$S_{coh}\left(f\right) = \gamma^2_{PV}\left(f\right) \cdot S_{tot}\left(f\right) \tag{10}$$

$$S_{dif}\left(f\right) = \left[1 - \gamma^2_{PV}\left(f\right)\right] \cdot S_{tot}\left(f\right) \tag{11}$$

In formulas (9) to (11), the total energy of the sonic field is represented as $S_{tot}\left(f\right) = S_{P^2}\left(f\right) + S_{V^2}\left(f\right)$. $S_{P^2}\left(f\right)$ and $S_{V^2}\left(f\right)$ express potential energy and kinetic energy respectively; $S_{coh}\left(f\right)$ denotes the coherent component of the total energy of the sound field. These components align with the linear relationship between the sound pressure signal and the particle velocity signal received by the vector hydrophone. This part of the energy is transmitted in a specific direction in the ocean. $S_{dif}\left(f\right)$ represents the diffused component of the total energy of the sound field that is not transmitted in the ocean. Therefore, it can be seen that the larger the value of the coherence function, the larger the proportion of the coherent component in the sound field, and vice versa.

## Theoretical simulation and experimental verification

### Theoretical simulation

Given that a wideband signal is incident on a single vector hydrophone, the incident azimuth angle $\theta = 20°$, the pitch angle $\phi = 0°$. Additional noise is white Gaussian noise independent of the signal. The SNR is 0 dB, and the sampling frequency is 20 kHz. Fig 1 presents sound pressure and particle velocity spectrum levels in three directions. Thus, the noise spectrum level $SL_{Vx}$ of Channel X is roughly equal to the sound pressure channel spectrum level $SL_p$, but it is approximately 8.5 dB and 15.3 dB higher than that of Y-channel $SL_{Vy}$ and Z-channel $SL_{Vz}$, respectively. Fig 2 displays the results of coherence analysis among the channels of the vector hydrophone. It can be seen that the coherence between the sound pressure channel and the X channel is about 0.95 across the entire frequency range, which is higher than the coherence (0.87) between the sound pressure channel and the Y channel. This is because the incident azimuth of the target is close to the direction of the X channel. As a result, the SNR of the received signal in the X-channel is slightly higher than that

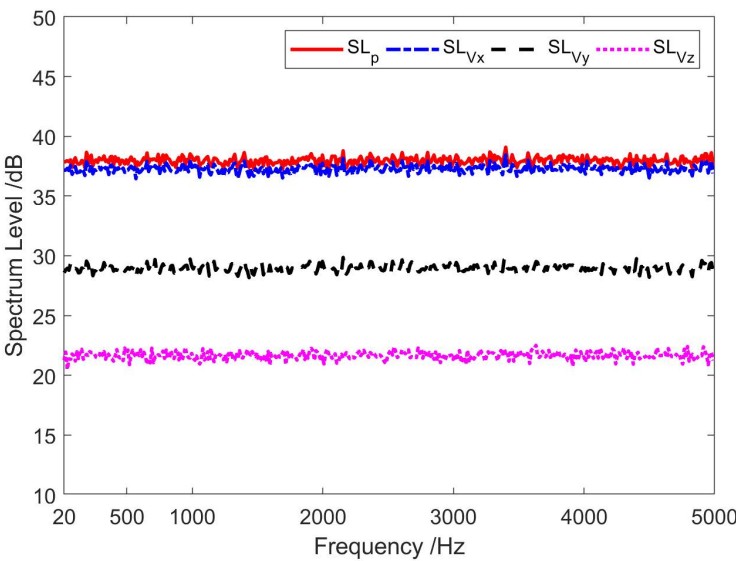

**Fig 1. Sound pressure and particle vibration velocity in three directions spectrum levels.**

**Fig 2. Coherence of sound pressure and particle vibration velocity signals.**

in the received signal of the Y-channel. Consequently, the coherence in the X direction is slightly better than that in the Y direction. The coherence between the sound pressure channel and the Z-channel is low, and the entire frequency range is mainly below 0.1. Because the target pitch angle is 0°, the component of the particle velocity in the Z-axis direction is very small. The Z-channel signal will be seriously interfered with by noise superposition caused by a significant proportion of the diffusion component.

The red solid line in Fig 2 represents the total coherence function of the sound field. The total coherence function is around 0.92 in the whole frequency range. Namely, the coherence component in the vector field accounts for about 92% of the total energy. In other words, the coherence spectrum is 0.36 dB lower than the spectrum of the total sound field, and the diffusion spectrum is 10.97 dB lower than the spectrum of the total sound field, as shown in Fig 3. Fig 4 and Fig 5 present the results of the energy component analysis of the sound field when SNR is −10 dB and −20 dB, respectively. In Fig 4, the coherence spectrum and the diffusion spectrum in the sound field are roughly the same when the SNR is −10 dB, and the overall coherence function is about 0.5. It can be inferred from Fig 5 that the energy component in the sonic field is mainly concentrated on the diffusion component when the SNR is −20 dB. On that basis, the overall coherence function is about 0.08, meaning that the diffusion component in the vector field accounts for about 92% of the total energy. Thus, the diffusion spectrum is 0.36 dB lower than the total sound field spectrum, and the coherence spectrum is 10.97 dB lower than the total sound field spectrum.

Fig 6 indicates the mean value of the coherence function between channels of a vector hydrophone in the frequency from 20 Hz to 5 kHz varying with SNR curve. The overall coherence function of the sound field, the coherence function between the sound pressure channel and the X-channel, and the coherence function between the sound pressure channel and the Y-channel all improve with the increase of the SNR. In addition, within the range of $\text{SNR} \in [-15, \; 0]$, the coherence function changes rapidly with the SNR. In the entire SNR range, the coherence function exhibits an "S"-shaped curve. When SNR>-15dB, the coherence function between the sound pressure channel and the X-channel is larger than the overall coherence function of the sound field, as well as the coherence function between the sound pressure channel and the Y-channel. It is consistent with the previous conclusion. Due to the target incident azimuth being close to the X-channel direction, the coherence function of the sound pressure channel and Z-channel does not significantly change with SNR. This is caused by the target incidence pitch angle of 0°.

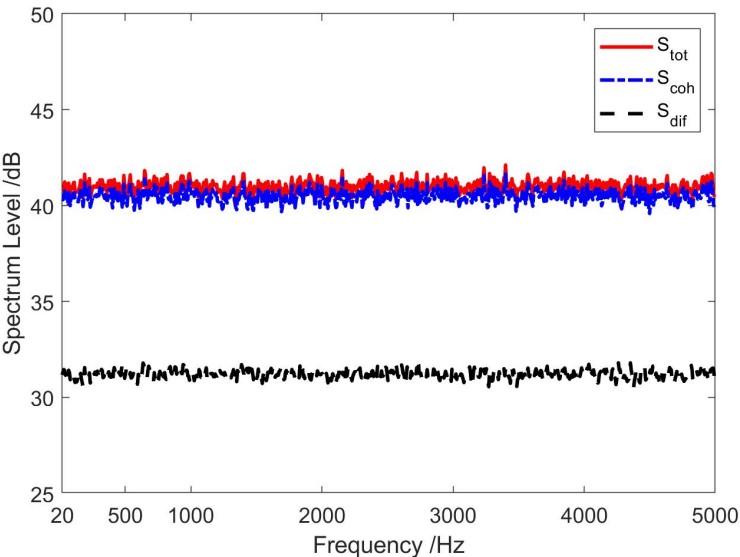

**Fig 3. Energy component analysis of the sound field (SNR = 0 dB).**

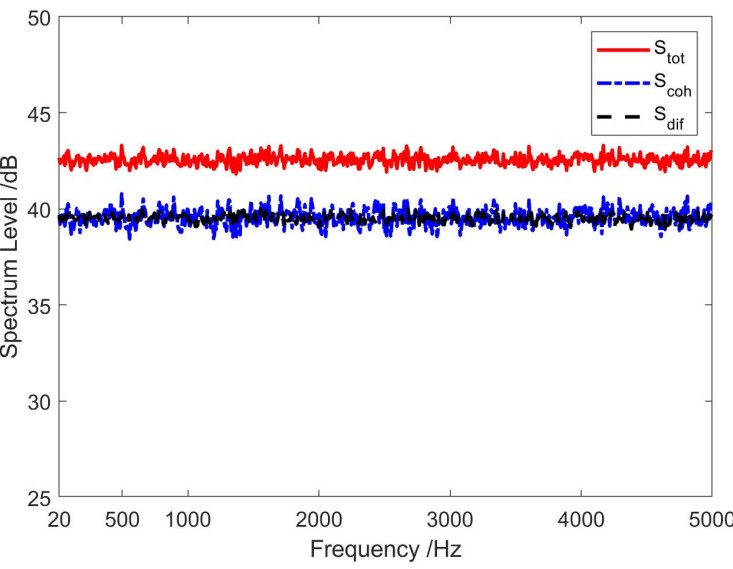

**Fig 4. Energy component analysis of the sound field (SNR = −10 dB).**

In order to investigate the influence of the target incidence azimuth and pitch angle on the coherence function of the sound field, Fig 7 and Fig 8 respectively illustrate the variation curves of the overall coherence function of the sound field with the target azimuth and pitch angle when the SNR is −15 dB, −10 dB, −4 dB, and 5 dB. It can be seen that the overall coherence function of the sound field basically does not change with the target incidence azimuth and pitch angle. Furthermore, the overall coherence function of the sound field is not related to the target incidence azimuth and pitch angle; its magnitude only depends on SNR of the received signal.

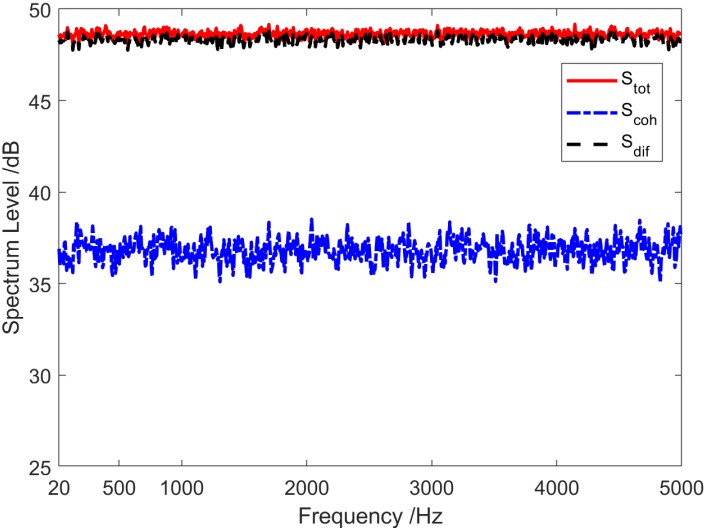

**Fig 5. Energy component analysis of the sound field (SNR=−20 dB).**

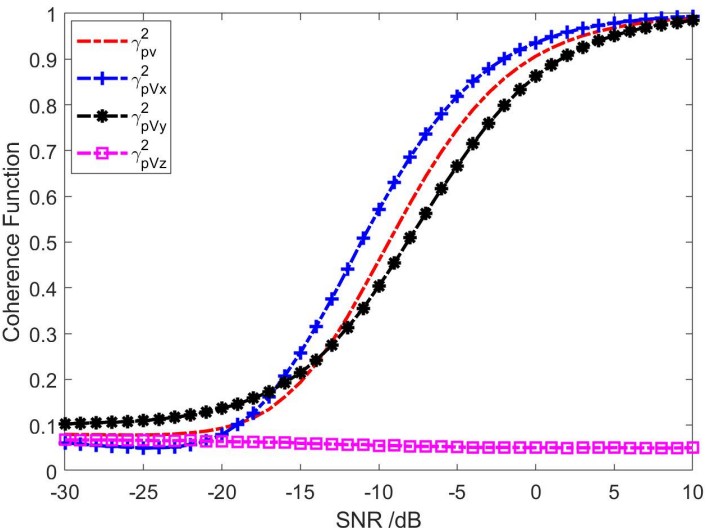

**Fig 6. The coherence curve of sound pressure and particle vibration velocity varying with SNR.**

The curve of each energy component varying with the SNR in the sound field is displayed in Fig 9. Initially, under the assumption that the SNR>-10dB, the energy component in the sound field is primarily the coherent component. Secondly, when SNR<-10dB, the energy component in the sound field is primarily the diffusion component. Fig 10 illustrates how the proportion of the coherent component and diffusion component in the sound field varies with SNR. In addition, Table 1 shows the proportion of each energy component in the sound field with nine SNRs. From Fig 10 and Table 1, it can be observed that as the increase of SNR, the proportion of coherent component in the sound field gradually uplifts, while the diffusion component decreases accordingly. Last, when SNR=-10dB, the coherent component in the vector field

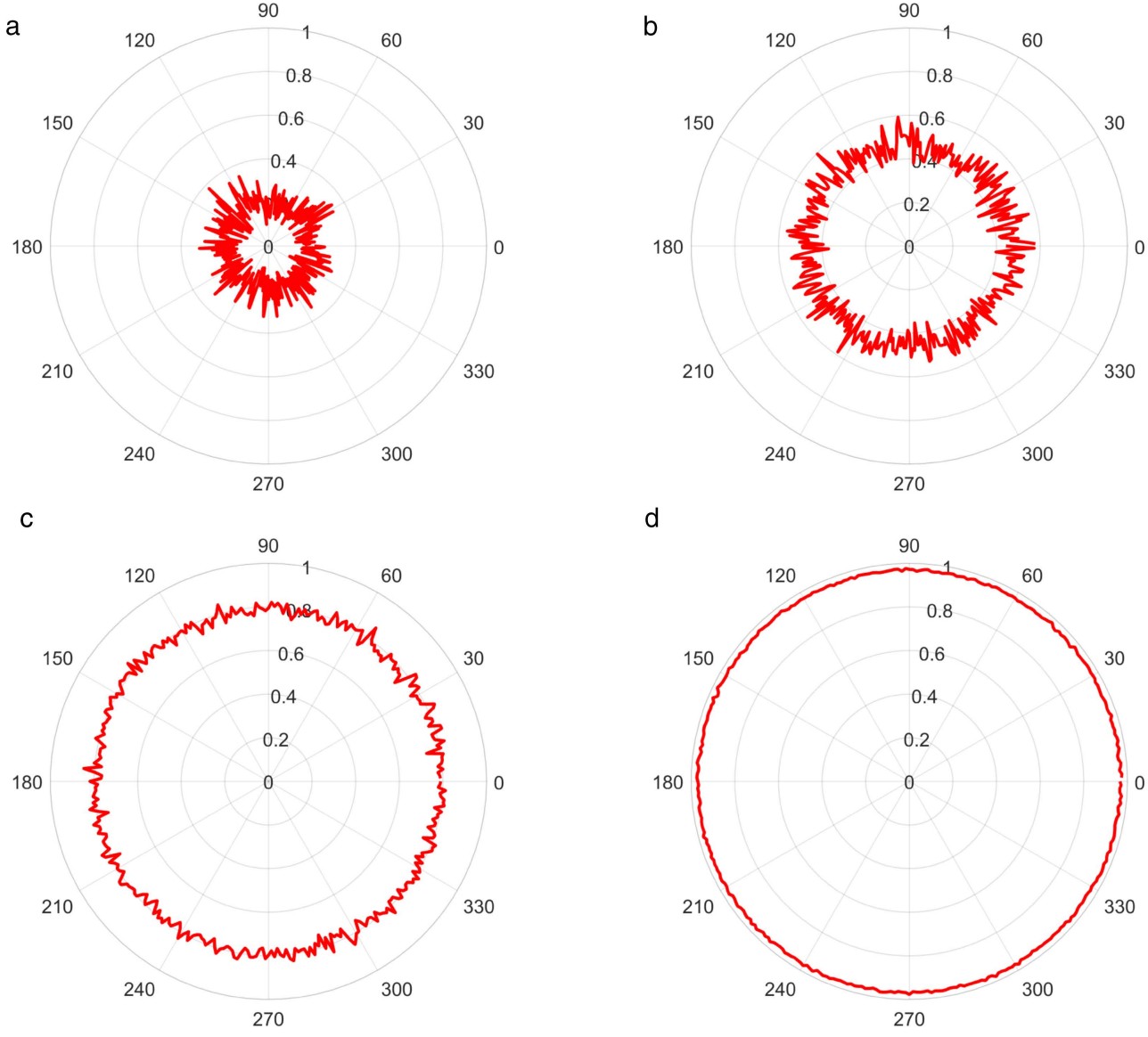

**Fig 7. The overall coherence function of the sound field varying with the target azimuth.** (a) SNR = -15dB, (b) SNR = -10dB, (c) SNR = -4dB, (d) SNR = 5dB.

accounts for about 52.1% of the total energy. Namely, the coherent spectrum is 2.8 dB lower than that of the total sound field, and the diffusion spectrum is 3.2 dB lower. In other words, the proportion of the coherent component in the sound field is basically the same as that of the diffuse component under these conditions. From Fig 6 to Fig 10, setting the threshold value of the overall coherence function or the threshold value of the proportion of the coherent component in the sound field enables the detection of the target.

Fig 11 represents the frequency characteristics of the phase difference between the sound pressure of vector hydro-phone and particle velocity in three directions under the simulation conditions consistent with Fig 1. The phase difference

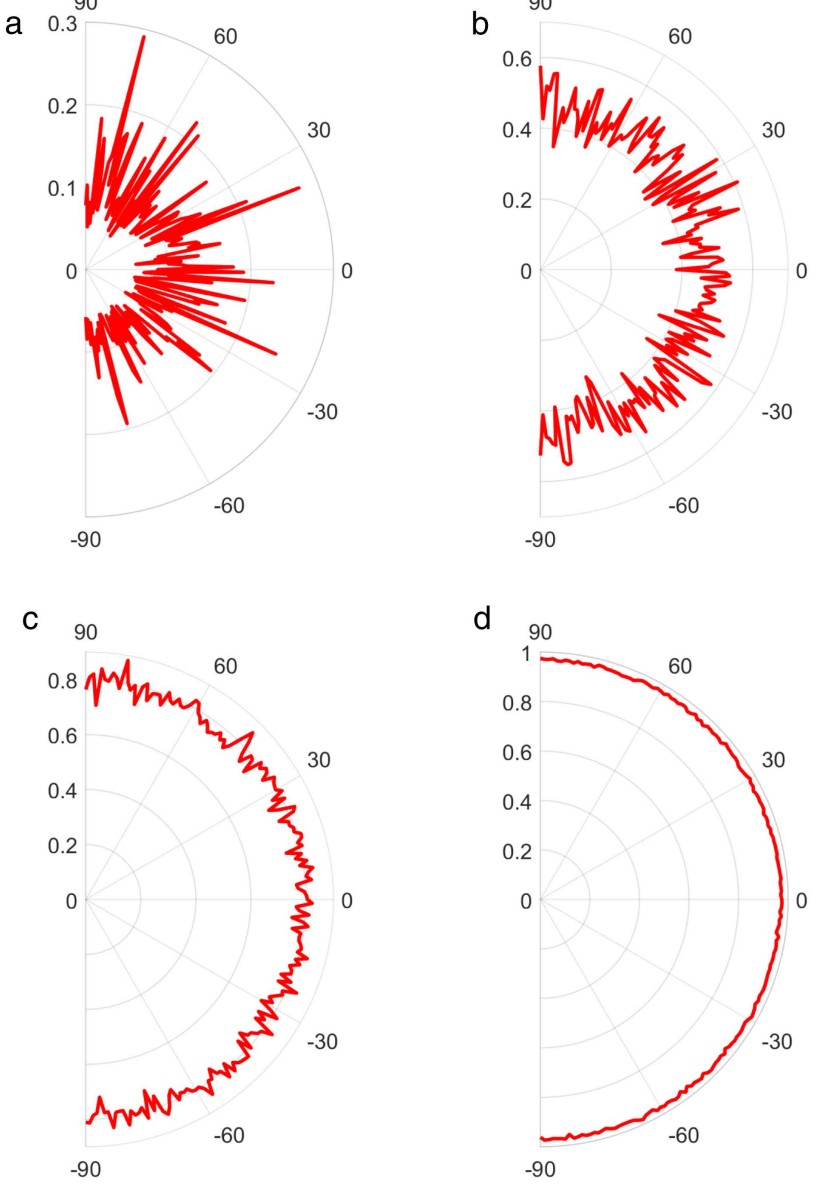

**Fig 8. The overall coherence function of the sound field varying with the pitch angle of the target.** (a) SNR = -15dB, (b) SNR = -10dB, (c) SNR = -4dB, (d) SNR = 5dB.

between sound pressure channel and X-channel, as well as between sound pressure channel and Y-channel are both about 0°. This indicates that the signal waveforms received by the sound pressure channel, the X-channel, and the Y-channel are approximately the same. Furthermore, they should exhibit a strong correlation. On the other hand, the phase difference between the sound pressure channel and the Z-channel is disorganized, showing irregular change between ±180°. This indicates a low correlation between the sound pressure channel and the Z-channel.

Fig 12 shows the curve of the phase difference with the target azimuth at a frequency of 1 kHz. The SNR adopted in the simulation is 0 dB. As exhibited in the graph, the phase difference between the P-channel and X-channel is near 0° in

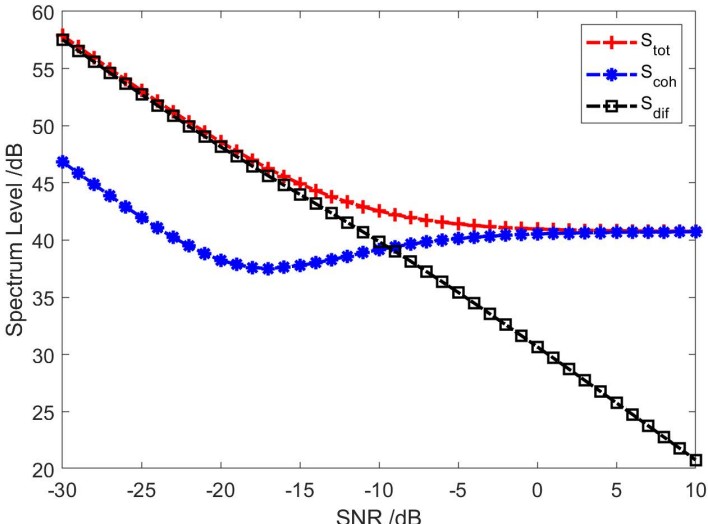

**Fig 9. The curve of each energy component in the sound field changing with SNR.**

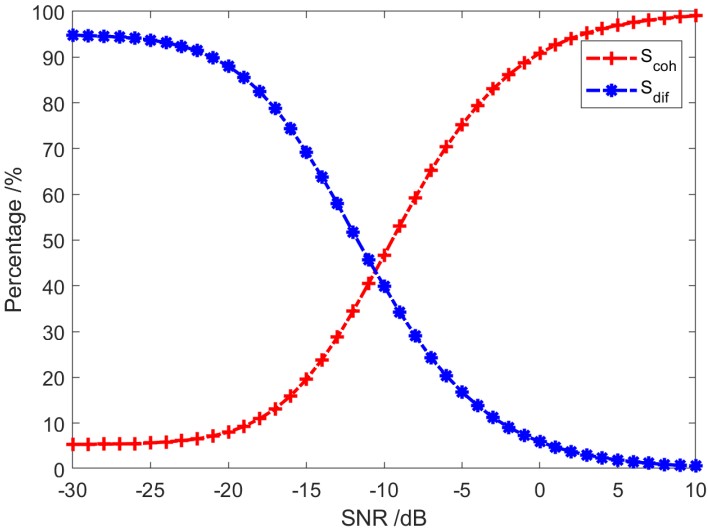

**Fig 10. The curve of the proportion of each energy component in the sound field varying with SNR.**

**Table 1. Proportion of each energy component in the sound field under various SNR conditions.**

| SNR(dB) | −20 | −18 | −16 | −14 | −10 | −6 | −4 | −2 | 0 |
|---|---|---|---|---|---|---|---|---|---|
| $S_{coh}$(%) | 19.2 | 21.4 | 25.6 | 31.9 | 52.1 | 73.3 | 81.4 | 87.5 | 91.7 |
| $S_{dif}$(%) | 80.8 | 78.6 | 74.4 | 68.1 | 47.9 | 26.7 | 18.6 | 12.5 | 8.3 |

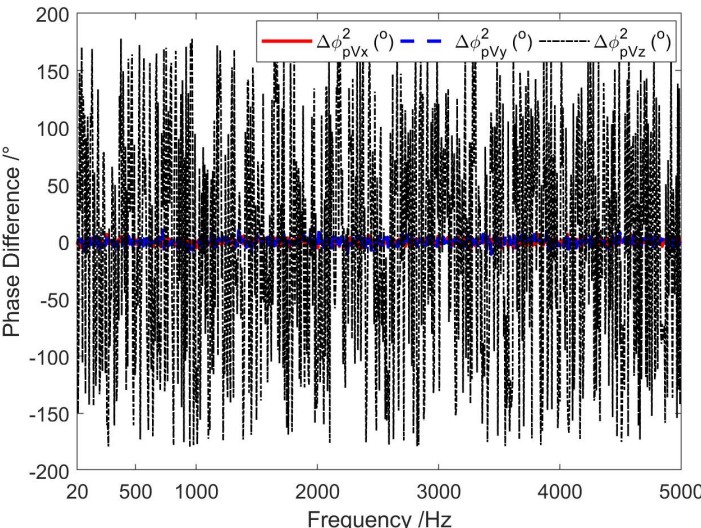

**Fig 11. Phase difference between sound pressure and particle vibration velocity in three directions.**

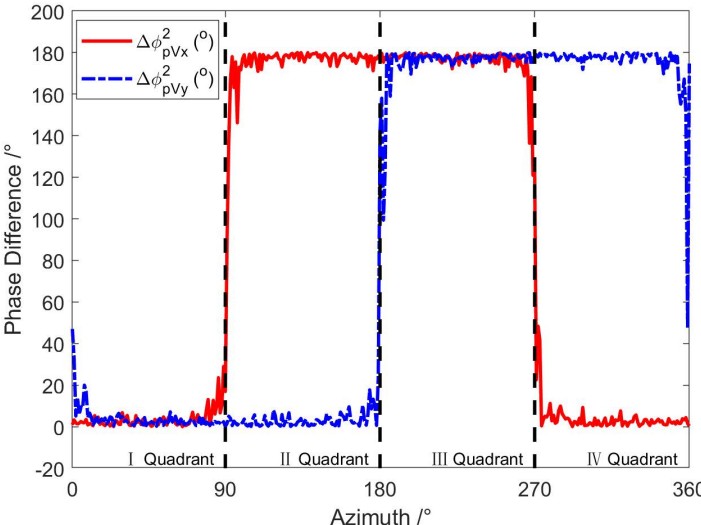

**Fig 12. The curve of phase spectrum $\Delta\varphi_{PV_x}$ and $\Delta\varphi_{PV_y}$ changing with target azimuth angle.**

the first and fourth quadrants, while it is close to 180° in the second and third quadrants. However, the phase difference between the P-channel and Y-channel is near 0° in the first and second quadrants, while it is close to 180° in the third and fourth quadrants. If SNR of the signal received by the vector hydrophone is infinite, then the phase difference between the P-channel and X-channel, as well as the P-channel and Y-channels will be strictly either 0° or 180°. Therefore, based on the phase difference, the quadrant where the target is located can be determined. If the phase difference results in a saltus, the incidence azimuth of the target can be determined.

The phase difference characteristics have been discussed above. According to the phase difference of each channel, the quadrant where the target is located can be judged. Similar to the phase difference characteristics, the quadrant

where the target orientation is located can also be determined according to the positive and negative acoustic energy flux in directions X and Y. Fig 13 is the curve of acoustic energy flow changing with the target azimuth angle at a frequency of 1 kHz, with the amplitude in the figure normalized. Since we only care about the acoustic energy flux being positive or negative, normalization is carried out in Fig 13. Namely, the sound energy flow is normalized to 1 when it is greater than 0 and normalized to −1 when it is less than 0. As shown in Fig 14, the quadrant where the target is positioned can also be identified based on the X and Y directions of the sound energy flow. If the symbol of the sound energy flow appears a saltus, the target azimuth can be determined.

## Anechoic pool testing

### Background noise treatment

In October 2023, a validation test was conducted in the anechoic pool of the National Deep Sea Center (NDSC) to advance the study of the conventional single-point coherence function and phase spectrum characteristics of sound fields. During the experiment, the vector hydrophone and UW350 sound source were set at a distance of 5m and both were located approximately 3m below the water surface, as shown in Fig 15.

Coherence results of the background noise tested in the anechoic pool are shown in Fig 16. Under the presence of background noise, the coherence between the channels of the vector hydrophone was low, except for specific frequency points, with values consistently below 0.2 across the entire frequency range. Fig 17 demonstrates the time-varying results of the background noise energy components measured in an anechoic tank. It is observed that on the condition of background noise, most of the energy components in the sound field were diffusion components. Fig 18 shows the frequency characteristics of the phase difference between the sound pressure of the vector hydrophone and the particle velocity in three directions. Therefore, in the background noise of an anechoic pool, the phase difference between the sound pressure channel and the three vector channels varied irregularly between ±180 °, suggesting that the sound pressure channel is not correlated with the three vector channels.

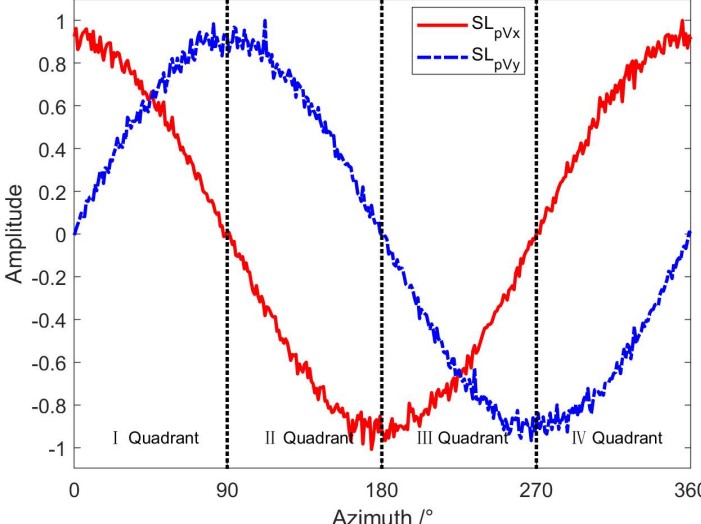

**Fig 13. The curve of sound energy flow in directions X and Y changing with the target azimuth angle.**

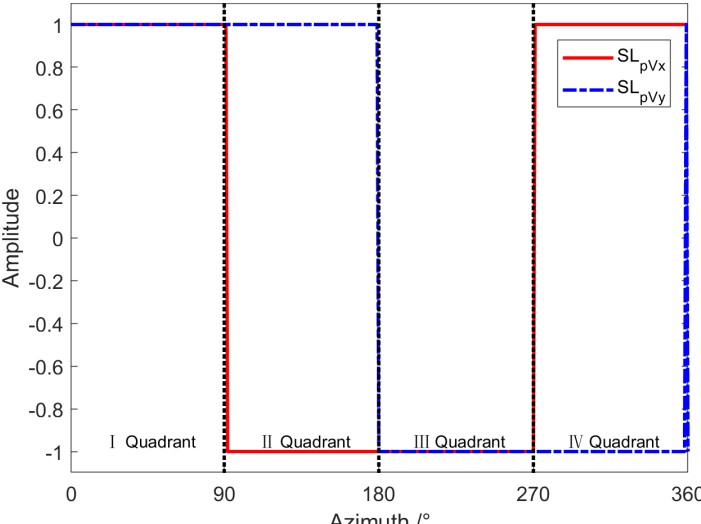

**Fig 14. The curve of acoustic energy flow changing with target azimuth angle after normalization.**

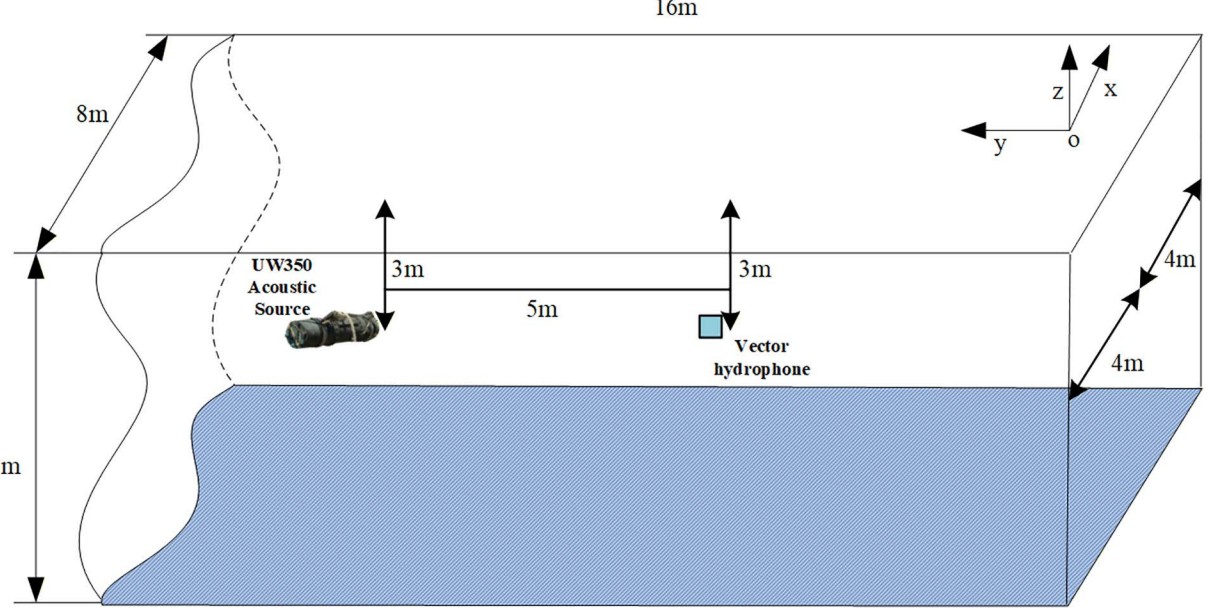

**Fig 15. Schematic diagram of the anechoic pool testing.**

## Single-frequency signal processing results

Fig 19 indicates coherence analysis results of the sound field when the sound source emits a 1 kHz single frequency pulse signal. The coherence function values were larger near the frequencies of 1 kHz, 2 kHz and 3 kHz. Among them, the pulse signals with frequencies of 2 kHz and 3 kHz were harmonic interference of the 1 kHz single-frequency signal emitted by the sound source. At the 1 kHz frequency point, the values of $\gamma^2_{PV_x}$, $\gamma^2_{PV_y}$ and $\gamma^2_{PV_z}$ were about 0.79, 1.0 and 0.95, respectively. This is due to the angle between the incident angle of the sound source signal and the direction of the X-channel,

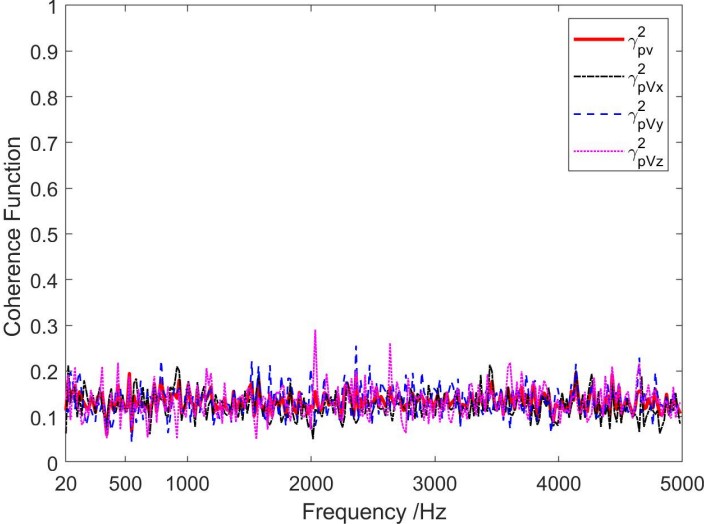

**Fig 16. Coherence of the background noise signal in an anechoic pool.**

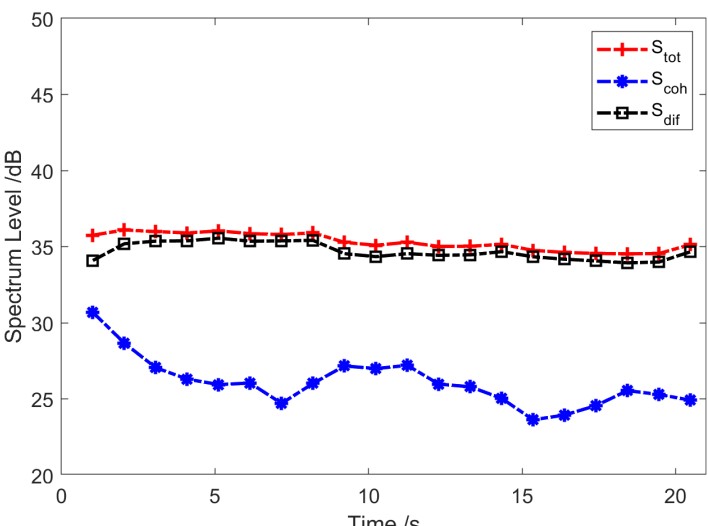

**Fig 17. Energy component analysis of background noise in an anechoic pool.**

Y-channel, and Z-channel directivity maximum being about 85°, 15° and 75°, respectively. As a result, the coherence between the sound pressure channel and the Y-channel is the largest, while the coherence between the sound pressure channel and the X-channel is relatively small.

Fig 20 presents the energy component analysis results of the sound field when the sound source emits a 1 kHz single-frequency pulse signal. At the frequency point of 1kHz, the coherence spectrum $S_{coh}(f_{1kHz})$ was essentially equal to the total sound field spectrum $S_{tot}(f_{1kHz})$, while the diffusion spectrum $S_{dif}(f_{1kHz})$ was approximately 25 dB lower than the total sound field spectrum $S_{tot}(f_{1kHz})$. In addition to the low frequency and the frequency range near 1 kHz and harmonic frequency, the coherence spectrum $S_{coh}(f_{1kHz})$ was, on average, 8 dB lower than the total sound field spectrum

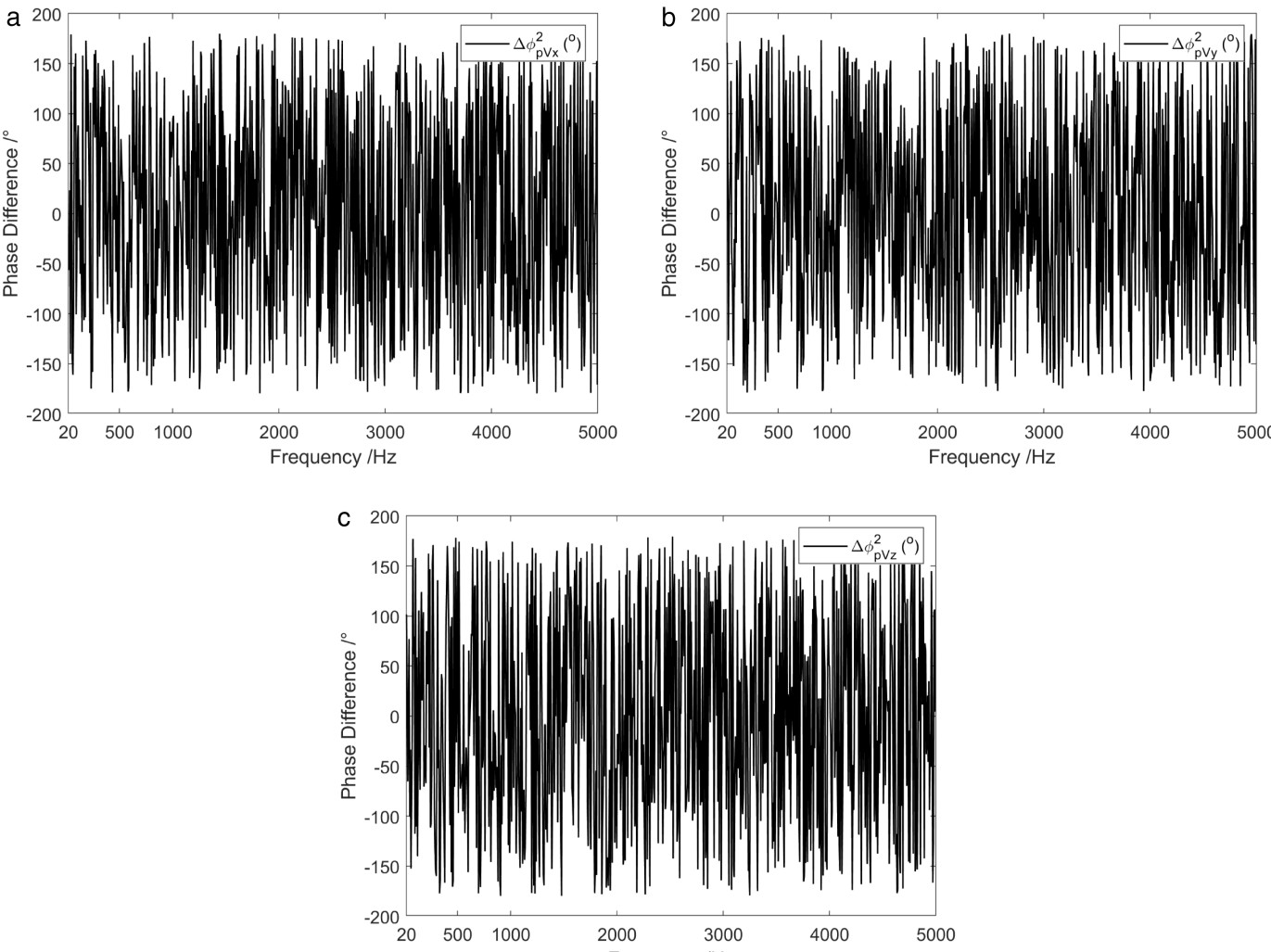

**Fig 18. Phase difference between sound pressure and particle velocity in three directions under background noise of anechoic pool.** (a) X-channel, (b) Y-channel, (c) Z-channel.

$S_{tot}(f_{1kHz})$. It can be seen from Fig 20 that target detection using coherence spectrum can achieve a higher SNR, which can enhance target detection performance.

The frequency characteristics of the phase difference between the sound pressure and the particle velocity in three directions when a sound source emits a 1 kHz single-frequency pulse signal are shown in Fig 21. In the vicinity of 1 kHz frequency point, the phase difference between sound pressure channel and X-channel, as well as between sound pressure channel and Z-channel were both about 0 °. Additionally, the phase difference between the sound pressure channel and the Y-channel was about 180°. In contrast, in other frequency bands other than 1 kHz, the frequency characteristics of the phase difference between the sound pressure channel and the three vector channels were chaotic. At the 1 kHz frequency point, the sound pressure signal and the vibration velocity signal waveforms were generally identical. This similarity results in higher coherence between the sound pressure signal at this frequency point and the three vibration velocity signals. The conclusion is consistent with that of Fig 19.

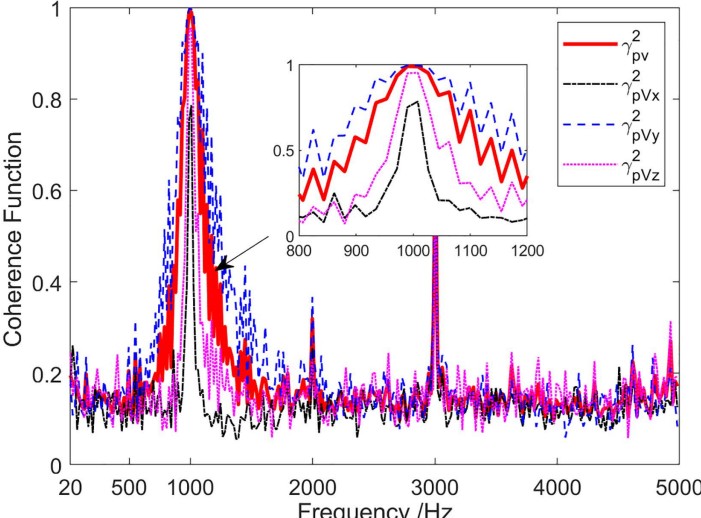

**Fig 19. Sound field coherence when the sound source transmits a 1 kHz single-frequency signal.**

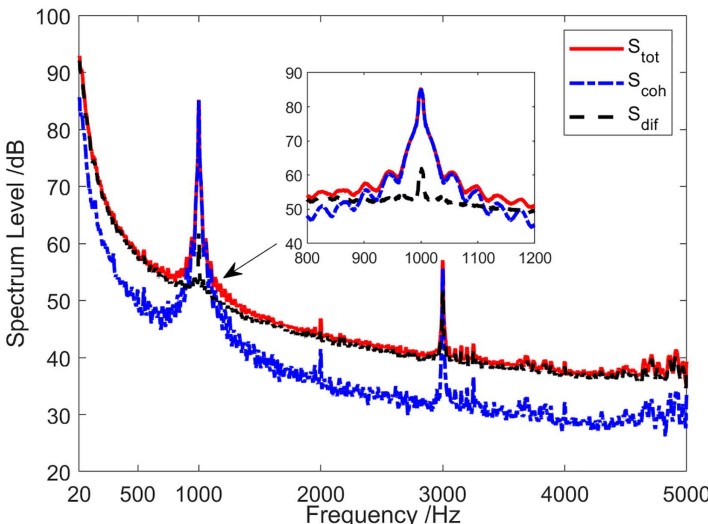

**Fig 20. Energy component analysis of sound field when the sound source transmits a 1 kHz single-frequency signal.**

## Broadband signal processing results

The analysis of sound field coherence when the sound source emits two types of broadband noise signals, low power and high power, is presented in Fig 22. In addition, it shows the curve of SNR of sound source signal varying with frequency. In the case of a low-power signal emitted by the sound source, the overall coherence function $\gamma^2_{PV} > 0.5$ of the sound field was only in the frequency range of SNR > 5.8 dB, and $\gamma^2_{PV_y} > \gamma^2_{PV_z} > \gamma^2_{PV_x}$ in this frequency range. Due to the angle between the incident angle of the sound source signal and the directivity maximum point direction of the X-channel, Y-channel, and Z-channel being about 90°, 0° and 58° respectively, the coherence between the sound pressure channel and the Y-channel was the highest, while the coherence between the sound pressure channel and the X-channel was relatively

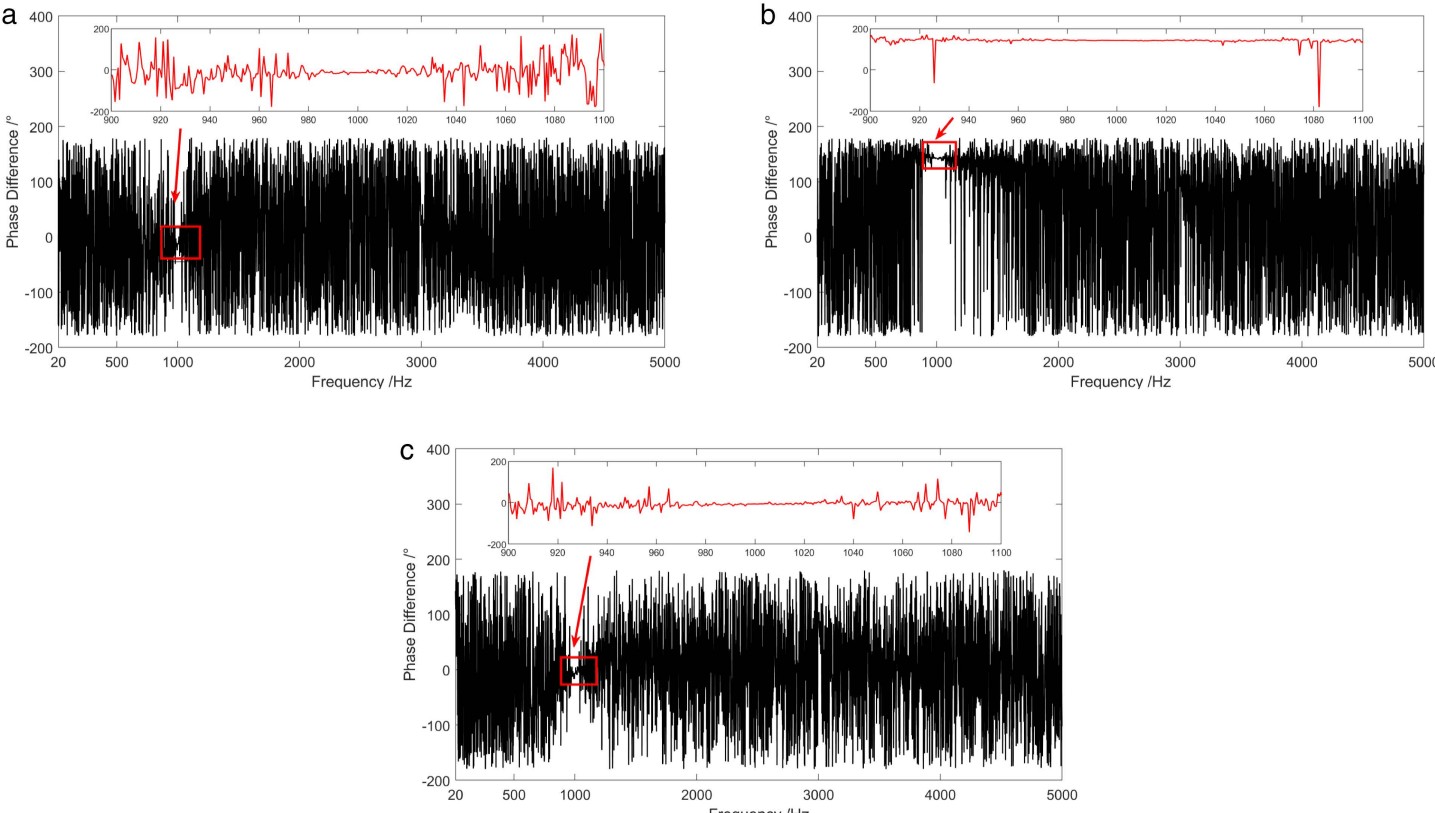

**Fig 21. Phase difference between sound pressure and particle vibration velocity in three directions when the sound source transmits 1 kHz single-frequency signal.** (a) X-channel, (b) Y-channel, (c) Z-channel.

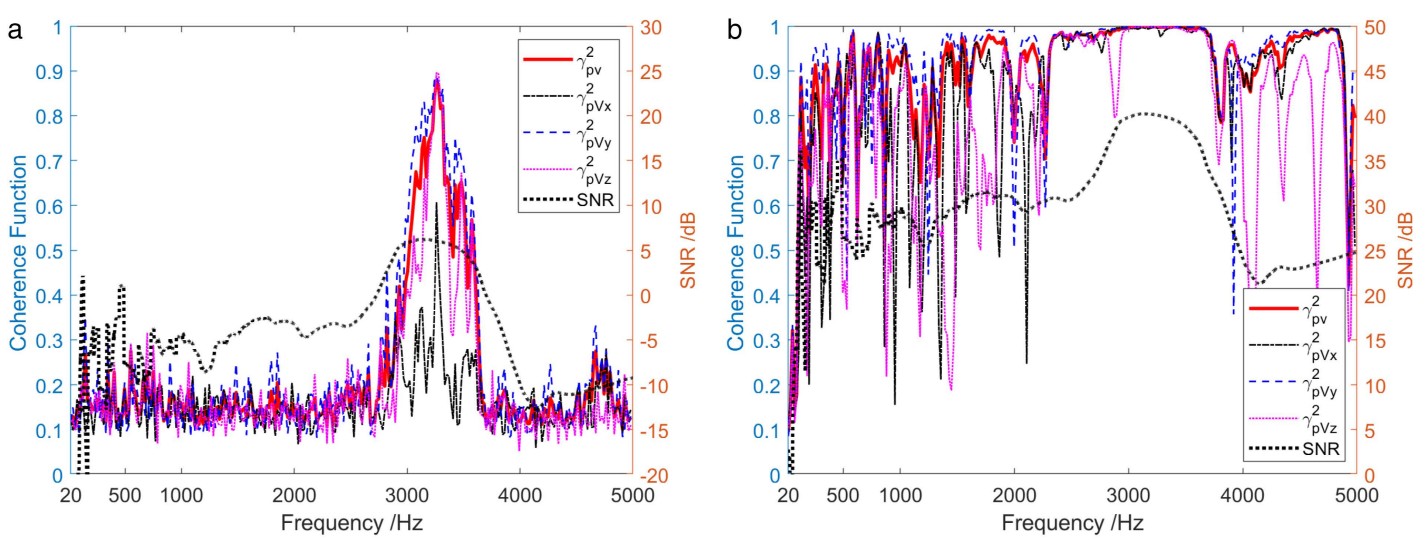

**Fig 22. Sound field coherence results when sound source transmits a broadband signal.** (a) Low power signal, (b) High power signal.

low. Furthermore, when a high-power signal is emitted by a sound source, except for specific frequency points, $\gamma^2_{PV_x}$, $\gamma^2_{PV_y}$ and $\gamma^2_{PV_z}$ values were larger among the entire frequency range.

Fig 23 depicts the analysis of sound field energy components when the sound source transmits low-power and high-power broadband noise signals. It can be obtained that when low-power signal is emitted by the sound source, only in the frequency range where SNR > 5.8 dB, the contribution of the coherent component in the sound field to the total sound field exceeded that of the diffusion field component. However, in the case of high-power signal emitted by the sound source, the coherence spectrum $S_{coh}$ was basically consistent with the total sound field spectrum $S_{tot}$ in the entire frequency range.

Fig 24 and Fig 25 display the frequency characteristic curve of the phase difference between the sound pressure and the particle velocity in three directions when the sound source emits low-power and high-power broadband signals, separately. First, when low-power signal is emitted by the sound source, the phase difference between the sound pressure channel and the three vector channels was relatively stable only in the frequency range of SNR > 5.8 dB. In the frequency range of low SNR, the phase difference frequency characteristics of the sound pressure channel and the three vector channels were disorganized. Secondly, in the case of high-power signal emitted by the sound source, the phase difference between the sound pressure channel and the three vector channels was relatively stable across the entire frequency range.

To further verify the advantages of the method proposed in this paper, we conducted comparative experiments using three methods under the same shallow water multi-path acoustic field environment and identical target and experimental parameters as shown in Table 2. The experimental parameters were: signal center frequency of 1kHz, signal-to-noise ratio (SNR) ranging from 5 to 15dB, false alarm probability (Pfa) of 10−3, and detection probability (Pd) and anti-multi-path interference gain were used as quantitative evaluation indicators (Ganti defined as the ratio of detection probability in a multi-path environment to that in a non-multi-path environment, with a value closer to 1 indicating stronger anti-interference capability).

Sound pressure-based spectrum detection: suitable for open seas without multipath interference, or scenarios demanding high real-time detection performance and limited hardware resources; Sound intensity-based detection: suitable for

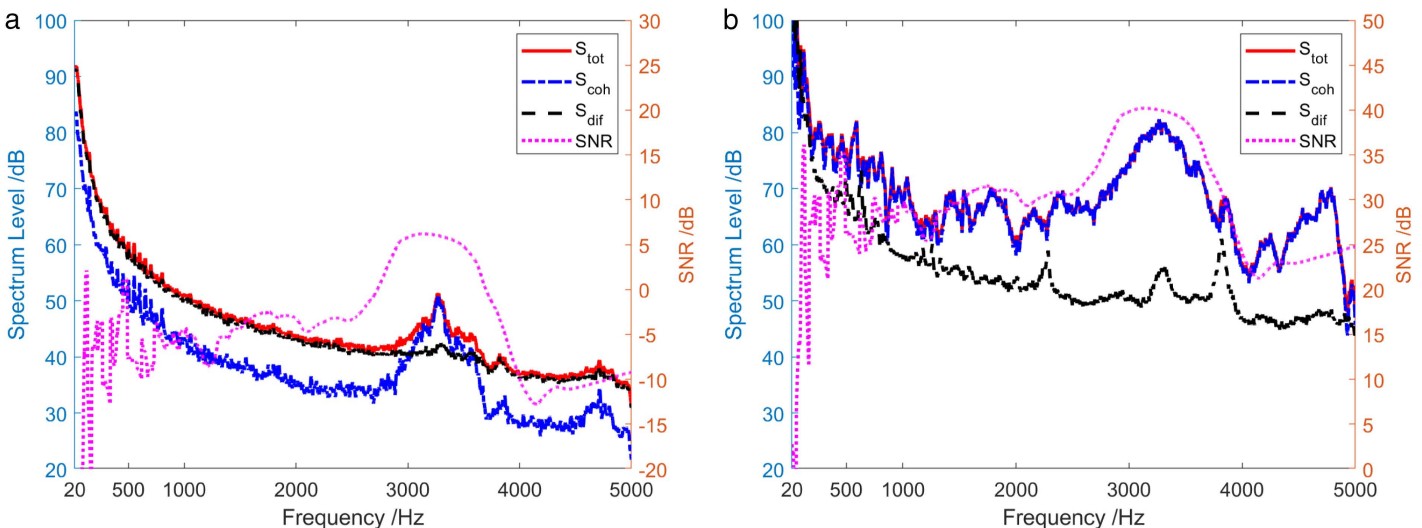

**Fig 23. Energy component analysis of the sound field when the sound source transmits a broadband signal.** (a) Low-power signal, (b) High-power signal.

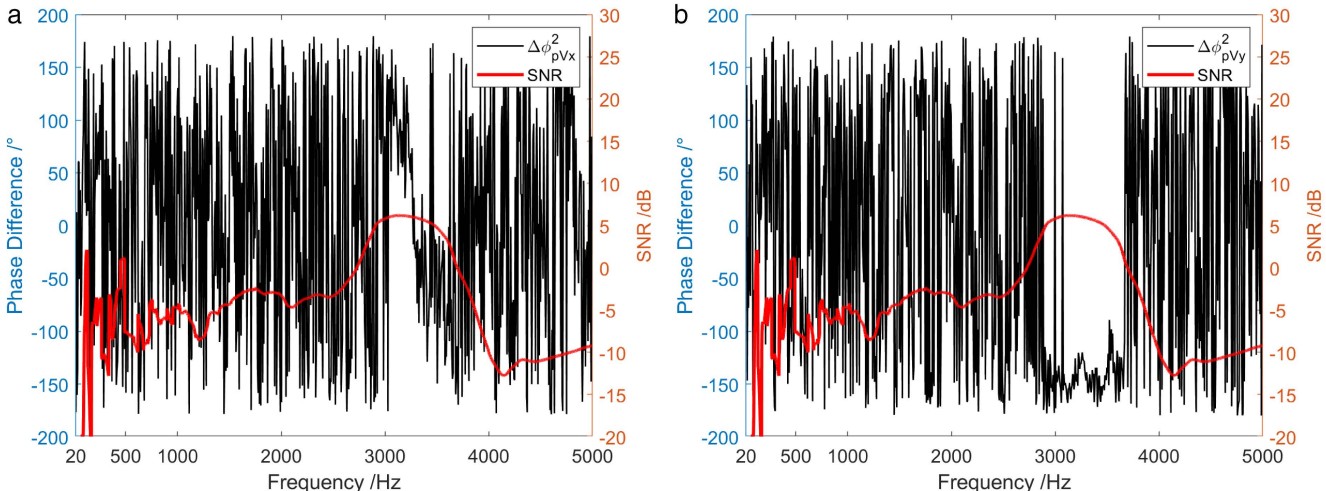

**Fig 24. Phase difference between sound pressure and particle vibration velocity in three directions when the sound source transmits a low-power broadband signal.** (a) X-channel, (b) Y-channel, (c) Z-channel.

scenarios with clear interference source directions, where sidelobe interference can be suppressed through beamforming; The detection method based on vector coherence in this paper: suitable for complex shallow water environments with severe multipath interference, particularly advantageous at low signal-to-noise ratios, and can serve as an important supplement to existing methods. The method in this paper explores the "hidden" coherence features between sound pressure and particle velocity components, providing a new feature dimension for target detection.

## Shallow sea testing

On October 11, 2023, the shallow sea verification test of the vector characteristics of the sound field of a single vector hydrophone was conducted off Qingdao. It was operated with wind speed level 2~3, sea state level 2, a sea depth of approximately 10m, and the sound velocity profile was uniform sound velocity distribution.

## Single-frequency signal processing results

It is shown in Fig 26 that the analysis of sound field coherence when the sound source emits a 1 kHz single-frequency pulse signal during the shallow sea testing. It can be observed that the coherence functions $\gamma^2_{PV_x}$, $\gamma^2_{PV_y}$ and $\gamma^2_{PV_z}$ were larger near the frequency of 1 kHz, namely 0.98, 0.98, and 0.92, respectively. The coherence function value was also relatively large in the high-frequency band of 3.4 kHz~5 kHz, which may be caused by existence of coherent interference in the test sea area within this frequency band.

The energy composition of the sound field when the sound source transmits a 1 kHz single-frequency pulse signal during the shallow sea testing is analyzed in Fig 27. At the frequency of 1 kHz, the coherence spectrum $S_{coh}(f_{1kHz})$ was essentially same as the total sound field spectrum $S_{tot}(f_{1kHz})$, and the diffusion spectrum $S_{dif}(f_{1kHz})$ was about 17 dB lower than the total sound field spectrum $S_{tot}(f_{1kHz})$. However, except for the low-frequency band of less than 100 Hz, the frequency band near 1 kHz and the high-frequency band of 3.4 kHz~5 kHz, the coherence spectrum $S_{coh}$ was about 8 dB lower than the total sound field spectrum $S_{tot}$ on average.

Fig 28 demonstrates frequency characteristics of the phase difference between sound pressure and particle velocity in three directions when a sound source emits a 1 kHz single frequency pulse signal during the shallow sea testing.

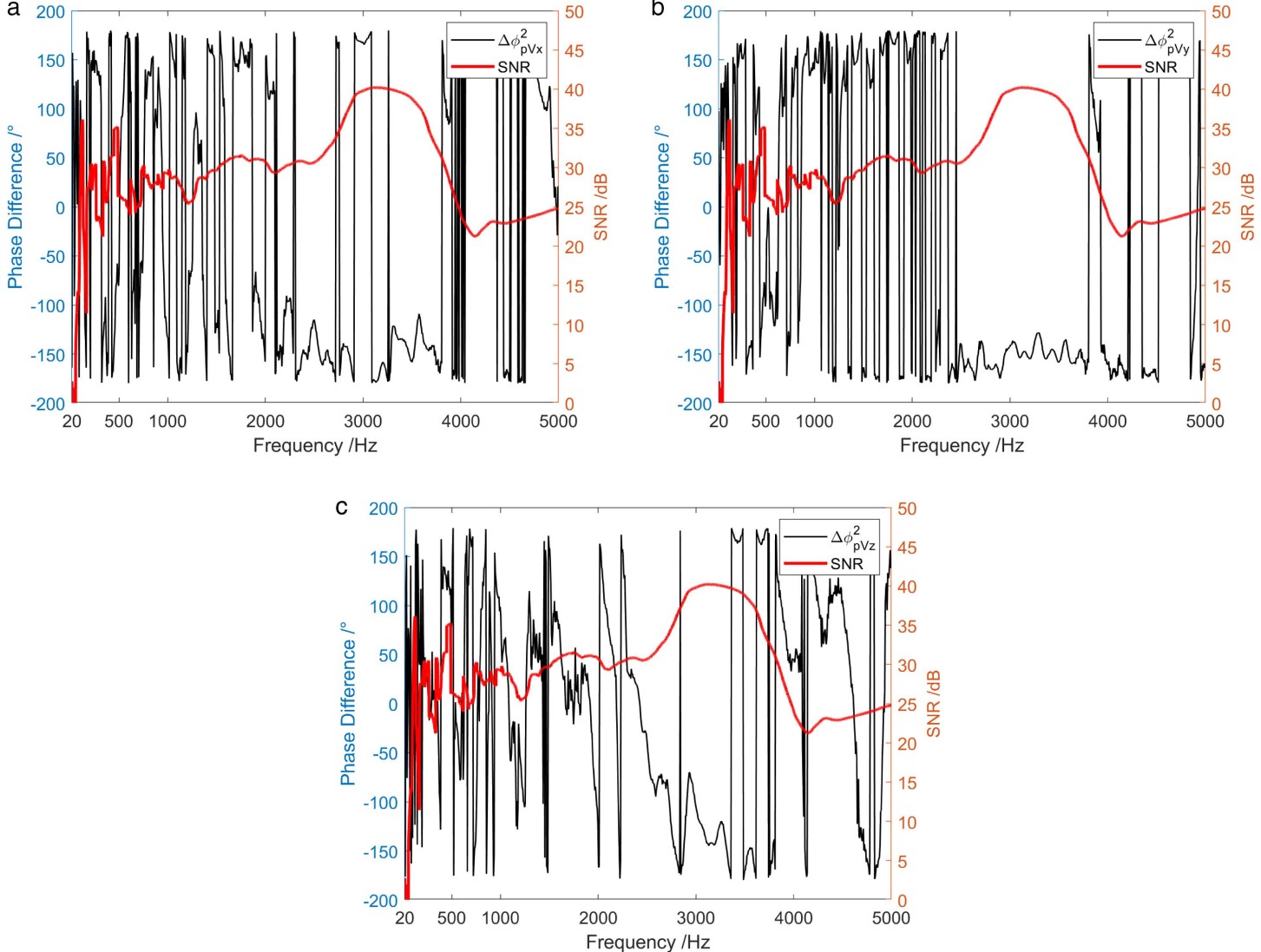

**Fig 25. Phase difference between sound pressure and particle vibration velocity in three directions when the sound source transmits a high-power broadband signal.** (a) X-channel, (b) Y-channel, (c) Z-channel.

Near the 1 kHz frequency point, the phase difference between the sound pressure channel and the X-channel, the Y-channel and the Z-channel was about 180°, 0°, and 0°, respectively. Moreover, the phase difference between the sound pressure and the particle velocity was relatively stable near the point. This stability attributes to the high coherence of the sound pressure signal and the vibration velocity signal near the 1 kHz frequency point. In the high frequency band of 3.4 kHz ~ 5 kHz, the phase difference between sound pressure and particle velocity in three directions also remained relatively stable attributed to coherent interference in this frequency band. It is consistent with the previous discussion results. Nevertheless, in addition to the frequency band near 1 kHz and the high frequency band of 3.4 kHz ~ 5 kHz, the phase difference frequency characteristics of the sound pressure channel and the three vector channels were chaotic.

**Table 2. Quantitative performance comparison of three methods.**

| SNR(dB) | evaluation metric | Spectral detection based on sound pressure | Detection based on sound intensity | This article is based on the detection of vector coherence | Compared to the sound pressure method; compared to the sound intensity method |
|---|---|---|---|---|---|
| 15 | Pd | 0.82±0.04 | 0.88±0.03 | 0.96±0.01 | 17.1% 9.1% |
| | Gain against multi-path interference(Ganti) | 0.65±0.05 | 0.78±0.04 | 0.92±0.02 | 41.5%;17.9% |
| 10 | Pd | 0.71±0.05 | 0.79±0.04 | 0.91±0.02 | 28.2%;15.2% |
| | Gain against multi-path interference(Ganti) | 0.52±0.06 | 0.69±0.05 | 0.86±0.03 | 65.4%;24.6% |
| 5 | Pd | 0.48±0.07 | 0.61±0.06 | 0.78±0.04 | 62.5%; 27.9% |
| | Gain against multi-path interference(Ganti) | 0.38±0.07 | 0.55±0.06 | 0.75±0.04 | 97.4%; 36.4% |

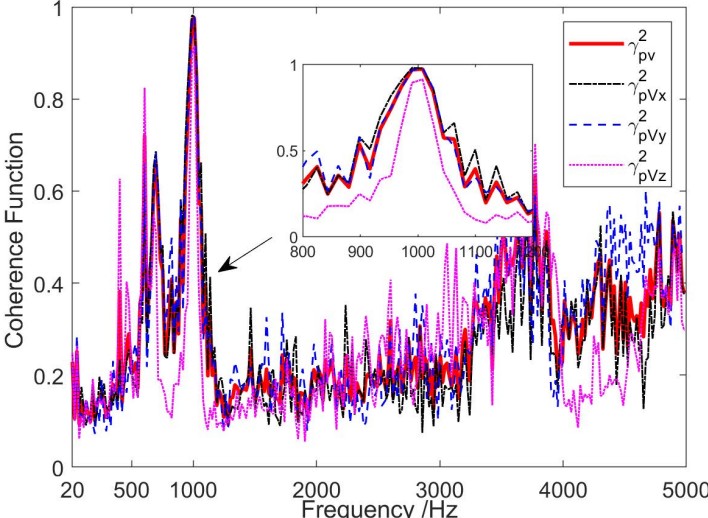

**Fig 26. Sound field coherence when the sound source transmits a 1 kHz single frequency signal.**

## Treatment of surface ship

From 1325 to 1334, the test ship (about 200 tons) passed near the location of the vector hydrophone at a speed of 7.7 kn and a heading of 258°. During this time period, the farthest and nearest distance between the test ship and the vector hydrophone were about 1.8 km (at 1325) and 0.05 km (at 1333), respectively. Fig 29 provides the relative situation diagram of the test ship and the vector hydrophone in this period.

Fig 30 shows the overall coherence function spectrum of the sound field in 1325~1334. First, when the target of the test ship passed near the vector hydrophone, the overall coherence function spectrum of the sound field presented obvious interference. Secondly, during 1325~1331 when the distance between the test ship and the vector hydrophone was relatively far, the coherence function was relatively stable and large in the high-frequency band of 3.4 kHz~5 kHz, and its value was greater than 0.3. While in the low-frequency band below 3.4 kHz, the value was basically below 0.2. This is probably because of the existence of 3.4 kHz~5 kHz interference noise sources in the test area. In the period of 1331~1334 when the distance between the test ship and the vector hydrophone is relatively close, the overall coherence

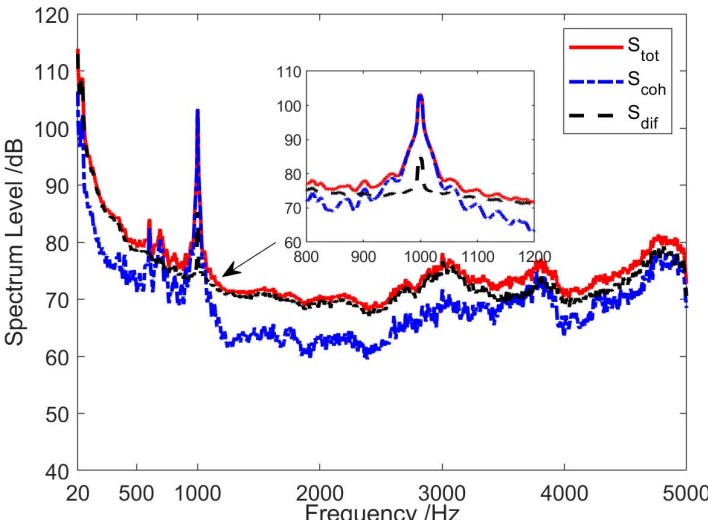

**Fig 27. Energy component analysis of the sound field when the sound source transmits a 1 kHz single-frequency signal.**

function increased rapidly. At 1331, it reached the maximum value (about 0.05 km away). The coherence function value at this time point was basically above 0.85 in the frequency range of 0.28 kHz~4.5 kHz.

Fig 31 provides the analysis results of sound field energy components in 1325~1334. When the test ship target passes near the vector hydrophone, the total sound field spectrum and the coherence spectrum exhibited notable interference characteristics. Furthermore, the coherence spectrum was stronger than the total sound field in spectrum interference characteristics, whereas the diffusion spectrum had no moving target interference. At 1325, the diffusion component accounted for about 88% of the total energy. This indicates that the diffusion spectrum is simply about 0.56 dB lower than the total sound field spectrum, and the coherence spectrum was about 9.2 dB lower than the total sound field spectrum. However, at 1333, the diffusion component accounted for only about 0.02% of the total energy. Specifically, the diffusion spectrum was about 16.99 dB lower than the total sound field spectrum, while the coherence spectrum was simply about 0.09 dB lower.

Fig 32 illustrates frequency spectrum results of the phase difference between the sound pressure and particle vibration velocity in three directions during 1325~1334. The phase difference spectra of the sound pressure channel and Z-channel showed obvious interference characteristics. Whereas, the phase difference spectra of the sound pressure channel and X-channel, as well as the sound pressure channel and Y-channel, showed no interference of moving targets. This is caused by the fact that the maximum direction of the Z-channel points to the sea surface, while the X-channel and Y-channel orients towards the water surface.

At the frequency below 3 kHz, the phase difference between the sound pressure channel and X-channel changed from 0° to 180° around 1333. This change is attributed to the target transitioning from the first quadrant of the vector hydrophone to the second quadrant. However, the phase difference between the acoustic pressure channel and the Y-channel was basically 0° around 1333 when the test ship is in close proximity to the vector hydrophone. The target in this time period was located in the first and second quadrants of the vector hydrophone. During 1325~1331 when the test ship was in a relatively long distance between the vector hydrophone, the phase difference of the sound pressure channel and X-channel, as well as the sound pressure channel and Y-channel were relatively stable, with values about 180° in the frequency range of 2.7 kHz~3.6 kHz. Thus, the interference signal in this frequency band was located in the third quadrant of the vector hydrophone. In the range of 4.0 kHz~4.8 kHz frequency band, the phase difference between the sound pressure channel and the X-channel was approximately 0°, while the phase difference between the sound pressure channel

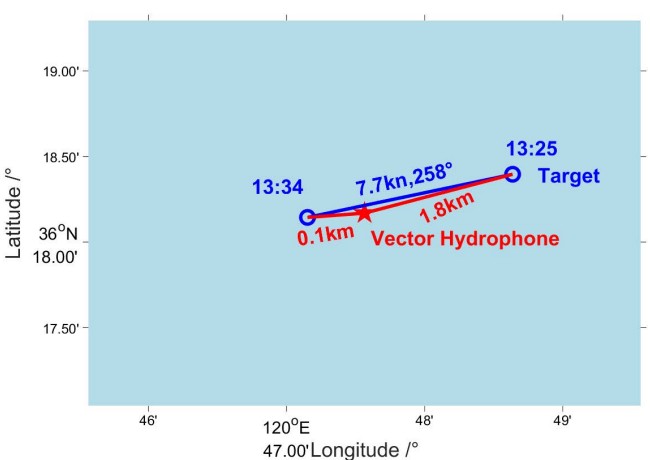

**Fig 28. Phase difference between the sound pressure and particle vibration velocity in three directions when the sound source transmits a 1 kHz single frequency signal.** (a) X-channel, (b) Y-channel, (c) Z-channel.

**Fig 29. Situation diagram of the test ship and vector hydrophone.**

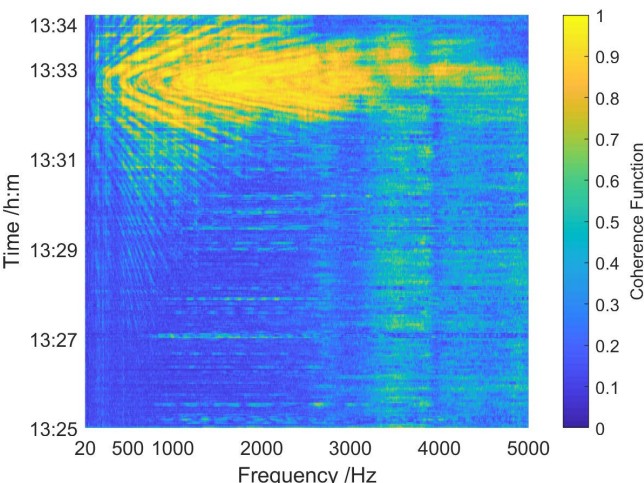

**Fig 30. Coherence spectrum of the sound field of the test ship in shallow sea.**

and the Y-channel was around 180°. This indicates that the interference signal in this frequency band was situated in the fourth quadrant of the vector hydrophone.

### Deep-sea testing

In November 2022, a deep-sea verification test was conducted to assess the vector characteristics of the sound field using a single vector hydrophone in a specific area of the South China Sea. The sea state during the test was approximately at level 4. From 1705 to 1759, the Dianke-1 test ship, weighing around 3,000 tons, approached the vector hydrophone's location at a speed of 7.6 kn following a course of 223°. Fig 33 shows the relative situation diagram of the test ship and the vector hydrophone during this period. Fig 34 is the curve of the heading angle, roll angle, and pitch angle of the vector hydrophone varying with time from 1705 to 1759. It can be observed that there is a periodic rotation phenomenon in the heading angle of the vector hydrophone, and the rotation period lasted about nine minutes. The pitch angle of the vector hydrophone fluctuated in a small range from 4° to 20°, while the roll angle remained relatively stable at about −6°.

The spectrum of the overall coherence function in 1705~1759 is offered in Fig 35. When Dianke-1 passed near the vector hydrophone, the overall coherence function spectrum of the sound field also appeared obvious interference characteristics. However, there was a "concave point" in the coherence function value near the 3.9 kHz frequency point in the entire time period. This issue may be caused by the vector hydrophone's poor response to the specific frequency point. From 1726 to 1734 when the distance between the test ship and the vector hydrophone was relatively close (less than 1 km), the overall coherence function was high across the whole frequency range with values generally exceeding 0.8. In cases where the distance between the test ship and vector hydrophone was relatively long (greater than 1 km), the overall coherence function value was typically below 0.3.

Fig 36 presents an analysis of the energy components of the sound field from 1705 to 1759. It can be obtained that the total sound field spectrum and coherence spectrum exhibited clear target interference characteristics when the test ship passes near the vector hydrophone, whereas the diffusion spectrum showed no moving target interference phenomenon. Moreover, The conclusion is consistent with that in shallow sea.

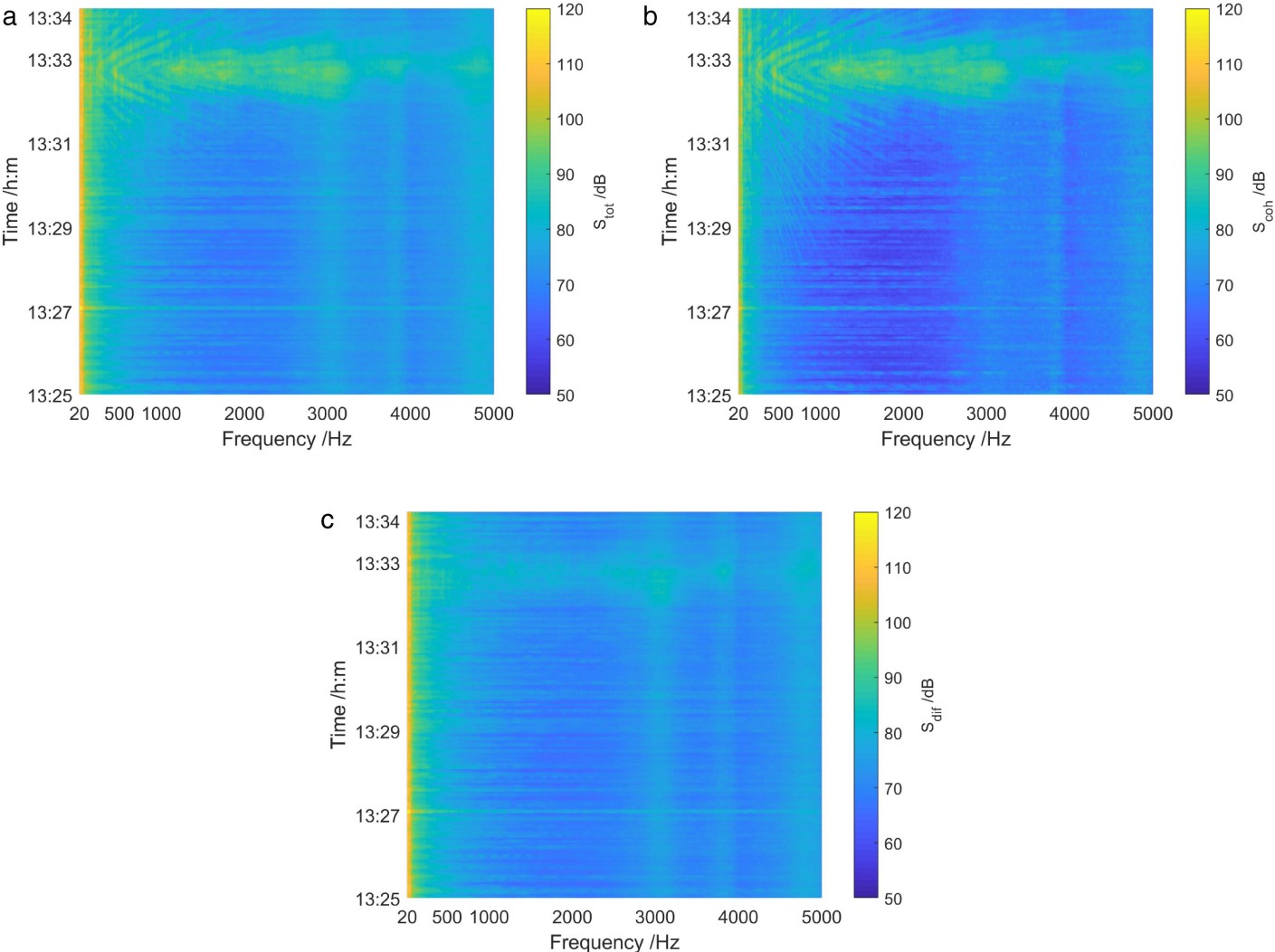

**Fig 31. Energy component analysis of the sound field of the test ship in shallow sea.** (a) The total sound field spectrum, (b) The coherence spectrum, (c) The diffusion spectrum.

Fig 37 demonstrates the spectrum of the phase difference between the sound pressure and the particle velocity in three directions during 1705~1759. It can be concluded that the phase difference between the sound pressure channel and the X-channel changed from 0° to 180° around 1728. This change is due to the vector hydrophone moving from the first quadrant to the second quadrant. The phase difference between the sound pressure channel and the Y-channel was 0°. At 1730, the phase difference between the sound pressure channel and Y-channel varied from 0° to 180°. This is because of the target movement from the first quadrant to the fourth quadrant resulting from the periodic rotation of the vector hydrophone. During the time when the test ship was relatively close to the vector hydrophone, the phase difference between the acoustic pressure channel and the Z-channel was essentially 0°. This is because the angle between the ship and the positive direction of vector hydrophone Z-channel was less than 90°.

a

b

c

**Fig 32. Spectrum diagram of the phase difference between the sound field pressure and particle vibration velocity in three directions of the shallow sea test ship.** (a) X-channel, (b) Y-channel, (c) Z-channel.

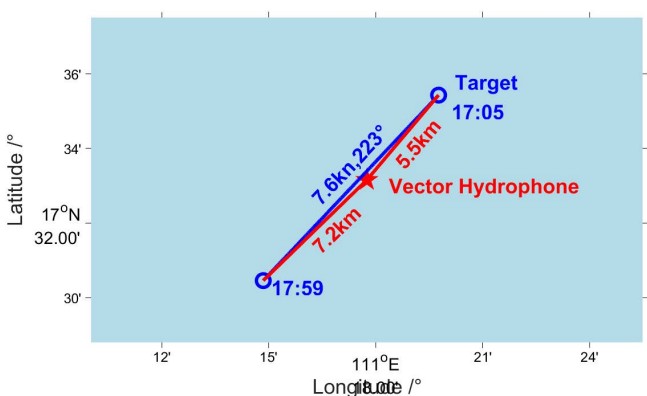

**Fig 33. Situation diagram of the test ship and vector hydrophone.**

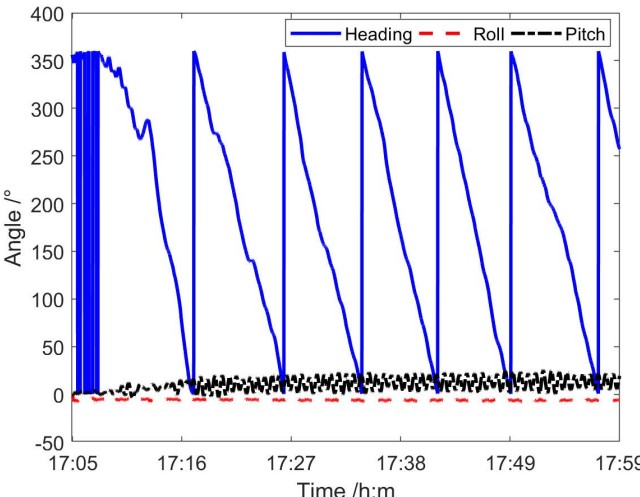

**Fig 34. Curve of vector hydrophone attitude angle varying with time.**

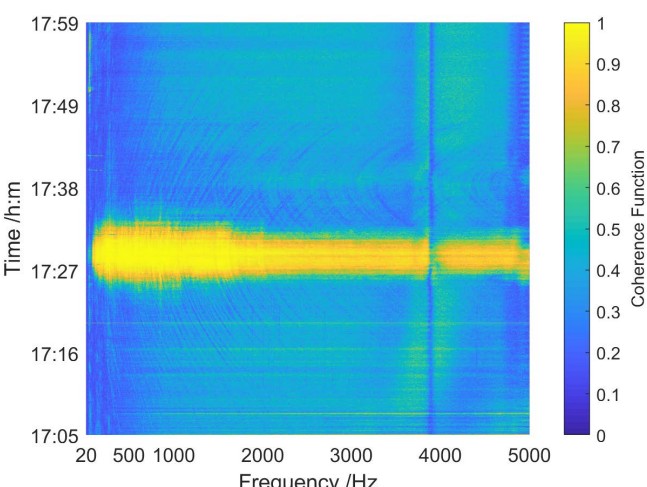

**Fig 35. Coherence spectrum of the sound field of the deep-sea test ship.**

## Conclusion

In this paper, three vector characteristics of the sound field received by a single vector hydrophone are discussed. The coherence function between the sound pressure of a vector hydrophone and the particle vibration velocity in three directions is quantitatively analyzed. Furthermore, the total sound field spectrum, coherence spectrum, diffusion spectrum, and phase spectrum of sound field are numerically analyzed and verified through experiments. The simulation and test results indicate as follows: 1) When the SNR is less than −10 dB, the energy component of the sound field is mainly concentrated in the diffusion component. However, when the SNR is greater than -10dB, the energy component is primarily the coherent component. Moreover, the overall coherence function of the sound field only depends on the received

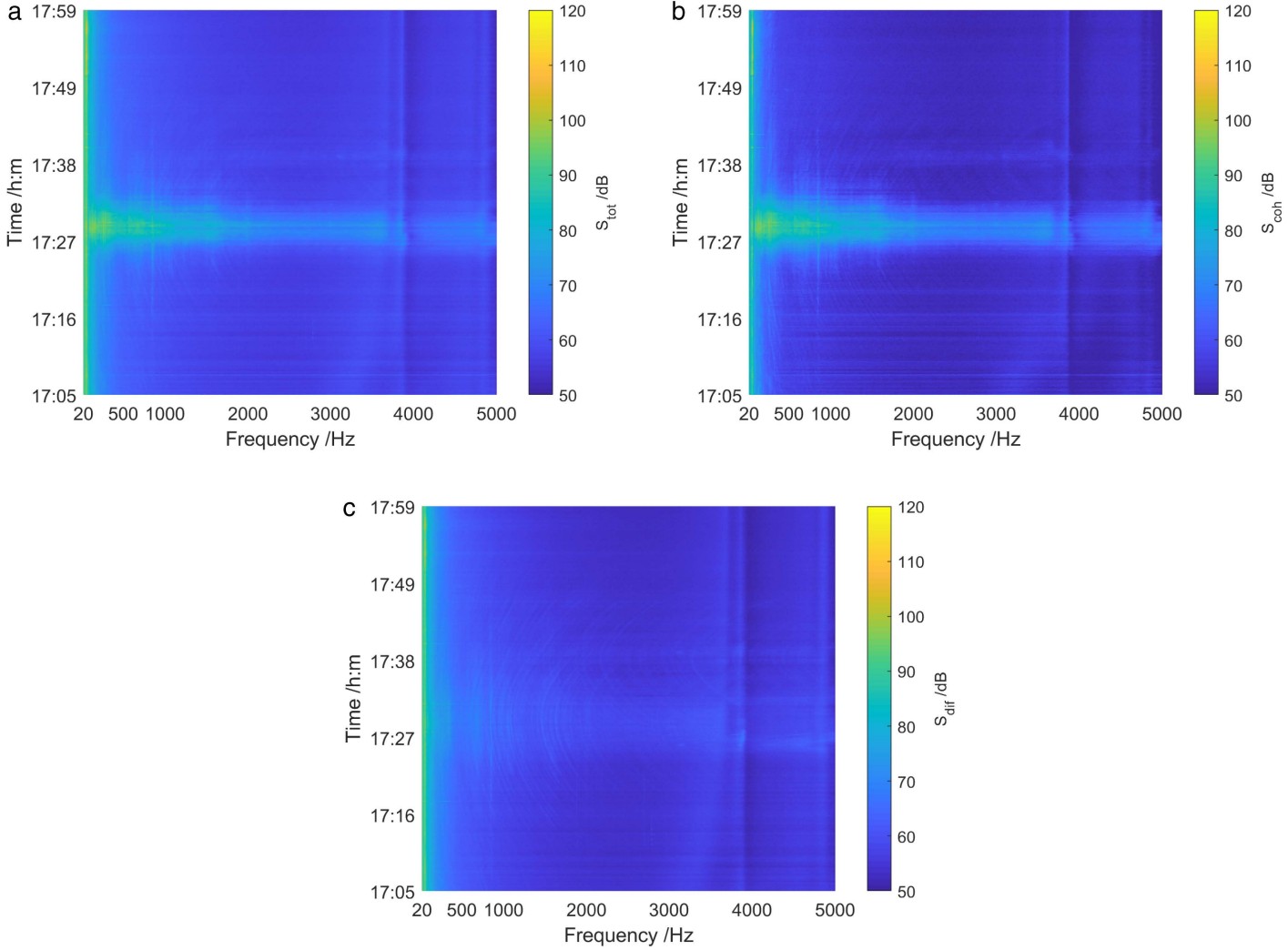

**Fig 36. Energy component analysis of sound field of deep-sea test ship.** (a) Total sound field spectrum, (b) Coherence spectrum, (c) Diffusion spectrum.

signal's SNR, and is unrelated to the target incident azimuth angle and pitch angle. 2) When the target is far away, the coherence between each channel of the vector hydrophone is small. 3) The coherence spectrum can improve the detection performance of the target, enhancing the SNR by about 8 dB compared to the total sound field spectrum. 4) The phase difference between sound pressure and particle vibration velocity in three directions can be utilized to determine the quadrant where the target is located. When the target SNR is low, the phase difference undergoes irregular changes. Accordingly, when the SNR is high, the phase difference remains relatively stable. 5) The overall coherence function of the sound field, the total sound field spectrum, and the coherence spectrum possess notable interference characteristics of the moving target enabling effective target detection. In conclusion, by studying the vector characteristics of the sound field, we can establish a foundation and generate ideas to enhance the practical engineering application of the vector hydrophone.

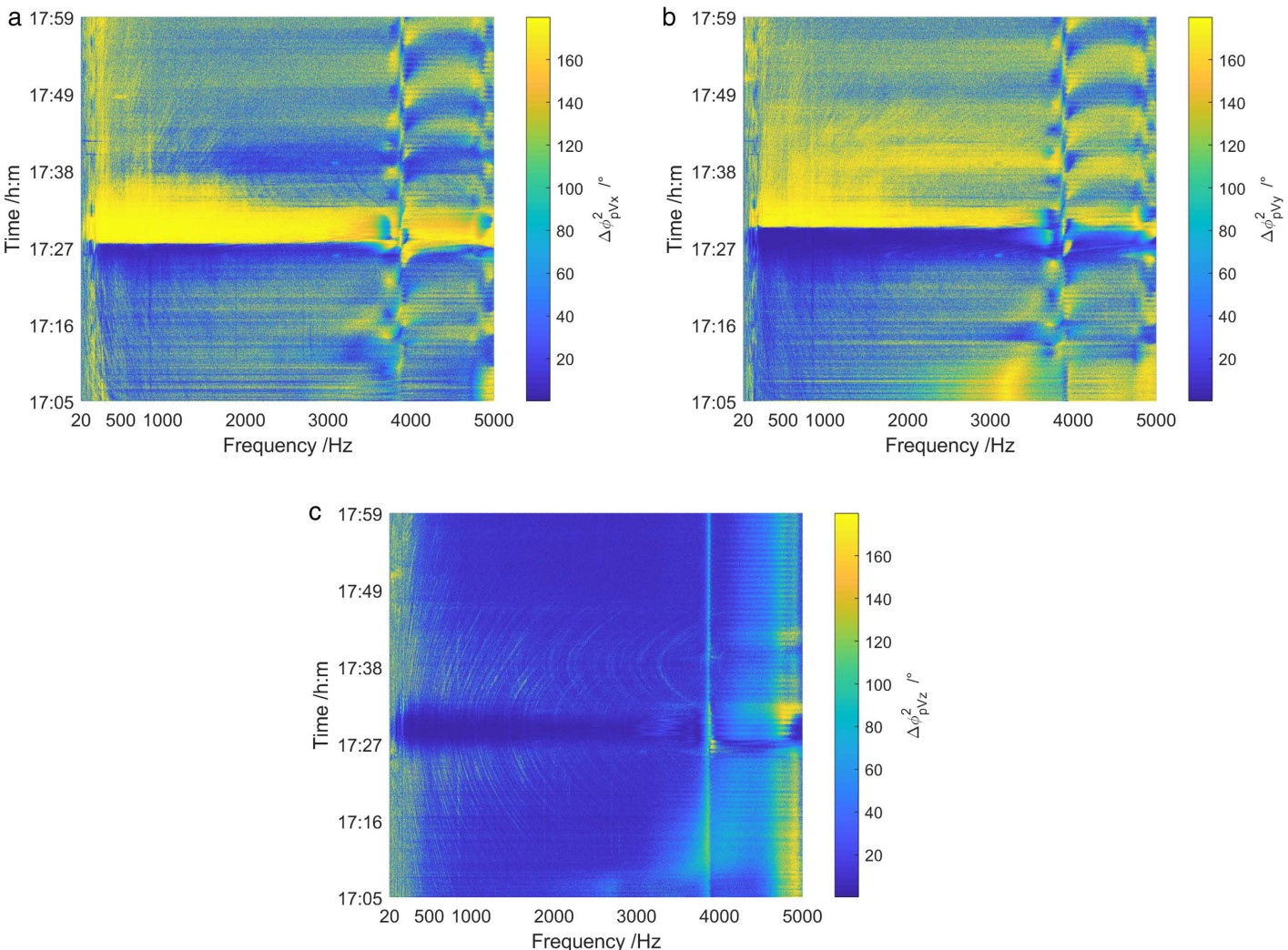

**Fig 37. Spectrum diagram of the phase difference between the sound field pressure and particle vibration velocity in three directions of the deep-sea test ship.** (a) X-channel, (b) Y-channel, (c) Z-channel.

## Author contributions

**Conceptualization:** Chao Wang.

**Data curation:** Chao Wang.

**Formal analysis:** Shuyang Jia.

**Funding acquisition:** Shuyang Jia.

**Investigation:** Yanhou Zhang.

**Methodology:** Yanhou Zhang.

**Project administration:** Qi Zhang.

**Resources:** Qi Zhang.

**Software:** Rongxin Zhu.

**Supervision:** Rongxin Zhu.

**Validation:** Rongxin Zhu.

**Visualization:** Rongxin Zhu.

**Writing – original draft:** Rongxin Zhu.

**Writing – review & editing:** Rongxin Zhu.

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
