## [Decision Letter · Decision Letter 0]

30 Dec 2025

PONE-D-25-60793Vector Characteristics of the Sound Field Based on a Single Vector HydrophonePLOS One

Dear Dr. Jia,

Thank you for submitting your manuscript to PLOS ONE. After careful consideration, we feel that it has merit but does not fully meet PLOS ONE’s publication criteria as it currently stands. Therefore, we invite you to submit a revised version of the manuscript that addresses the points raised during the review process.

We look forward to receiving your revised manuscript.

Kind regards,

Yufeng Zhou, PhD

Academic Editor

PLOS One

Journal Requirements:

“National Key R&D Program of China ( Grant No. 2021YFC3100900 ), Laoshan Laboratory Science and Technology Innovation Project ( Grant No. LSKJ202201100 ), and The Innovation Plan of Qingdao Institute of Collaborative Innovation ( Grant No. LYY-2022-05 ).”

Reviewers' comments:

Reviewer's Responses to Questions

**Comments to the Author**

1. Is the manuscript technically sound, and do the data support the conclusions?

Reviewer #1: Yes

Reviewer #2: Yes

2. Has the statistical analysis been performed appropriately and rigorously? 

Reviewer #1: Yes

Reviewer #2: Yes

3. Have the authors made all data underlying the findings in their manuscript fully available?

Reviewer #1: No

Reviewer #2: Yes

4. Is the manuscript presented in an intelligible fashion and written in standard English?

Reviewer #1: Yes

Reviewer #2: Yes

5. Review Comments to the Author

Reviewer #1: The manuscript is closely related to underwater acoustics and vector sensor signal processing. It presents a large amount of experimental data and provides a relatively complete physical interpretation of coherence-based vector features. However, the work is essentially descriptive and phenomenological, and several theoretical assumptions, methodological details, and quantitative justifications still require clarification or strengthening before it meets publication standards.

Major Concerns

1. Lack of mathematical rigor in key assumptions

Several critical assumptions are not sufficiently justified: the energy decomposition formula implicitly assumes statistical independence between coherent and diffuse components; interpreting the coherence function directly as an energy proportion relies on certain prerequisites. These assumptions are neither explicitly stated nor discussed in terms of their applicability limits.

2. Insufficient quantitative evaluation

Most conclusions are based on qualitative trends shown in the figures, lacking quantitative metrics to substantiate the claimed performance improvement. At least one quantitative detection performance metric should be added to support the assertion of “performance enhancement.”

3. Unclear relationship to existing detection methods

Although the manuscript claims that vector features can be used for target detection, it does not compare the proposed coherence-based method with established approaches (e.g., pressure-only spectral detection or intensity-based detection).

Minor Comment

4. While generally readable, the English expression contains repetitive phrasing, and the discussion section could be further refined. Professional language editing is recommended.

Reviewer #2: The manuscript entitled “Vector Characteristics of the Sound Field Based on a Single Vector Hydrophone” presents a comprehensive investigation of vector acoustic field characteristics obtained from a single vector hydrophone. The authors systematically analyze coherence functions, energy component proportions, and phase spectrum characteristics, supported by theoretical simulations as well as anechoic pool, shallow-sea, and deep-sea experimental validations.

I believe the manuscript is basically sound and publishable, and only minor revisions are required to further improve clarity, rigor, and presentation quality.

1. The manuscript defines an “overall coherence function” by summing coherence contributions from three orthogonal velocity components. What is the physical justification for this aggregation? It would be better to add some explanations about the Eq.4.

2. On page 6, the simulations assume additive white Gaussian noise independent of the signal. How sensitive are the reported coherence thresholds and energy decomposition results to this assumption? Would correlated or directional ambient noise significantly alter the conclusions?

3. Coherence values are presented as smooth curves averaged over frequency in Fig.6. What is the temporal averaging length used in the estimation? How stable are these coherence estimates under shorter observation times?

4. In shallow-sea experiments, multipath propagation and boundary reflections are unavoidable. How do these effects influence the coherence function and phase spectrum, and how are they distinguished from target-induced coherence?

5. The conclusions suggest broad applicability of coherence and phase characteristics for target detection. What are the identified limitations of the proposed approach in terms of frequency band, source type, or environmental complexity?

6. PLOS authors have the option to publish the peer review history of their article (what does this mean?). If published, this will include your full peer review and any attached files.

Reviewer #1: No

Reviewer #2: No

---

## [Author Response · Author response to Decision Letter 1]

30 Jan 2026

We sincerely appreciate your valuable comments, please see Response to reviewers file.

---

## [Decision Letter · Decision Letter 1]

18 Feb 2026

Vector Characteristics of the Sound Field Based on a Single Vector Hydrophone

PONE-D-25-60793R1

Dear Dr. Jia,

We’re pleased to inform you that your manuscript has been judged scientifically suitable for publication and will be formally accepted for publication once it meets all outstanding technical requirements.

Kind regards,

Yufeng Zhou, PhD

Academic Editor

PLOS One

Additional Editor Comments (optional):

Reviewers' comments:

Reviewer's Responses to Questions

**Comments to the Author**

1. If the authors have adequately addressed your comments raised in a previous round of review and you feel that this manuscript is now acceptable for publication, you may indicate that here to bypass the “Comments to the Author” section, enter your conflict of interest statement in the “Confidential to Editor” section, and submit your "Accept" recommendation.

Reviewer #1: All comments have been addressed

Reviewer #2: All comments have been addressed

2. Is the manuscript technically sound, and do the data support the conclusions?

Reviewer #1: Yes

Reviewer #2: Yes

3. Has the statistical analysis been performed appropriately and rigorously? 

Reviewer #1: (No Response)

Reviewer #2: Yes

4. Have the authors made all data underlying the findings in their manuscript fully available?

Reviewer #1: Yes

Reviewer #2: Yes

5. Is the manuscript presented in an intelligible fashion and written in standard English?

Reviewer #1: Yes

Reviewer #2: Yes

6. Review Comments to the Author

Reviewer #1: (No Response)

Reviewer #2: (No Response)

7. PLOS authors have the option to publish the peer review history of their article (what does this mean?). If published, this will include your full peer review and any attached files.

Reviewer #1: No

Reviewer #2: No

---

## [Editor Report · Acceptance letter]

PONE-D-25-60793R1

PLOS One

Dear Dr. Jia,

I'm pleased to inform you that your manuscript has been deemed suitable for publication in PLOS One. Congratulations! Your manuscript is now being handed over to our production team.

Kind regards,

on behalf of

Dr. Yufeng Zhou

Academic Editor

PLOS One